# Mouse pulmonary interstitial macrophages mediate the pro-tumorigenic effects of IL-9

Yongyao Fu[1], Abigail Pajulas [1], Jocelyn Wang [1], Baohua Zhou [2], Anthony Cannon[1], Cherry Cheuk Lam Cheung[1], Jilu Zhang[1], Huaxin Zhou[3], Amanda Jo Fisher[3], David T. Omstead[4], Sabrina Khan [4], Lei Han[2], Jean-Christophe Renauld[5], Sophie Paczesny[6], Hongyu Gao[7], Yunlong Liu [7], Lei Yang[2], Robert M. Tighe [8], Paula Licona-Limón[9], Richard A. Flavell [10], Shogo Takatsuka[11], Daisuke Kitamura [11], Jie Sun [12], Basar Bilgicer[4], Catherine R. Sears [3], Kai Yang[2] & Mark H. Kaplan [1✉]

Although IL-9 has potent anti-tumor activity in adoptive cell transfer therapy, some models suggest that it can promote tumor growth. Here, we show that IL-9 signaling is associated with poor outcomes in patients with various forms of lung cancer, and is required for lung tumor growth in multiple mouse models. CD4$^+$ T cell-derived IL-9 promotes the expansion of both CD11c$^+$ and CD11c$^-$ interstitial macrophage populations in lung tumor models. Mechanistically, the IL-9/macrophage axis requires arginase 1 (Arg1) to mediate tumor growth. Indeed, adoptive transfer of Arg1$^+$ but not Arg1$^-$ lung macrophages to $Il9r^{-/-}$ mice promotes tumor growth. Moreover, targeting IL-9 signaling using macrophage-specific nanoparticles restricts lung tumor growth in mice. Lastly, elevated expression of IL-9R and Arg1 in tumor lesions is associated with poor prognosis in lung cancer patients. Thus, our study suggests the IL-9/macrophage/Arg1 axis is a potential therapeutic target for lung cancer therapy.

[1] Department of Microbiology and Immunology, Indiana University School of Medicine, Indianapolis, IN 46202, USA. [2] Department of Pediatrics and Herman B Wells Center for Pediatric Research, Indiana University School of Medicine, Indianapolis, IN 46202, USA. [3] Division of Pulmonary, Critical Care, Sleep and Occupational Medicine/Department of Medicine, Indiana University School of Medicine, Indianapolis, IN 46202, USA. [4] Department of Chemical and Biomolecular Engineering, University of Notre Dame, Notre Dame, IN 46556, USA. [5] Ludwig Institute for Cancer Research, Experimental Medicine Unit, Université Catholique de Louvain, Brussels 1200, Belgium. [6] Department of Microbiology and Immunology, Medical University of South Carolina, 173 Ashley Avenue, Charleston, SC 29425, USA. [7] Department of Medical and Molecular Genetics, Indiana University School of Medicine, Indianapolis, IN 46202, USA. [8] Division of Pulmonary, Allergy, and Critical Care Medicine, Duke University Medical Center, Durham, NC 27710, USA. [9] Departamento de Biologia Celular y del Desarrollo, Instituto de Fisiologia Celular, Universidad Nacional Autónoma de México, 04510 Mexico City, Mexico. [10] Department of Immunobiology, Yale University School of Medicine, New Haven, CT 06510, USA. [11] Research Institute for Biomedical Sciences (RIBS), Tokyo University of Science, Noda, Japan. [12] Department of Medicine, Mayo Clinic, Rochester, MN 55905, USA. ✉email: mkaplan2@iupui.edu

The recent demonstration of potent anti-tumor activity from T helper 9 (Th9) and other IL-9-producing cells supported these cells as an attractive strategy for cancer cell therapy[1,2]. Adoptive transfer of Th9 cells or other IL-9-producing cells limits tumor growth, particularly in models of melanoma[3–7]. IL-9 mediates anti-tumor effects through multiple mechanisms including impacting the function of T cells, mast cells and dendritic cells (DCs), and by directly killing tumor cells[5,6,8,9]. However, endogenous IL-9 signaling has pro-tumorigenic effects in multiple cancers[2]. $Il9^{-/-}$ mice showed dramatically diminished lung tumor growth when intravenously injected with 4T1, CT26 or TUBO cancer cell lines, indicating a tumorigenic role in the lung environment[10]. Mechanistically, a recent study showed that Th9-derived IL-9 directly promoted both human and mouse lung cancer cell migration and proliferation[11]. This finding is consistent with increased expression of $IL9R$ in patient tumor samples, and increased serum IL-9 level that correlates with progressive breast cancer[10,12,13]. However, understanding of how IL-9 might contribute to tumor growth and which cells are important in the tumor microenvironment for mediating those effects is still very limited.

Tumor-associated macrophages (TAMs) are one of the most abundant tumor-infiltrating immune cells[14]. The lung is particularly vulnerable to metastasis as it encompasses a large and dense vascular area, and lung macrophages have attracted great attention in the development of lung cancer. Alveolar macrophages (AM) and interstitial macrophages (IM) are the two major heterogeneous lung macrophage populations[15]. AMs reside in the alveoli as a frontline defense to the environment, and are derived from fetal liver and embryonic monocytes, maintaining a high level of self-proliferation[15]. IMs are located in the lung parenchyma[15]. Although both AM and IM populations can contribute to TAM populations[16,17], the functional diversity of TAMs is affected by stimuli from the tumor microenvironment. TAMs can prevent tumor growth by secreting cytokines and chemokines that activate the anti-tumor response[14]. In contrast, TAMs exhibiting an immunosuppressive phenotype promote tumor growth by directly inducing tumor angiogenesis, EMT, invasion and proliferation, or secreting inhibitory factors that suppress the immune response[14]. Thus, TAMs are potential therapeutic targets for lung cancer. Previous studies have shown that IL-9 affects the oxidative burst of human blood monocytes and AMs but little is known about how IL-9 affects the phenotype and plasticity of lung macrophages or whether this might be linked to tumor progression[18,19].

In this study, we find lung macrophages occupy a large proportion of IL-9-responsive cells in the tumor microenvironment. IL-9 alters macrophage subsets in the lung and promotes tumor growth in a macrophage- and Arg1-dependent manner. Evidence of this pathway in lung cancer patients is also presented. Thus, our study advances the current understanding of IL-9-mediated disease development and provides a rationale for therapeutically targeting the IL-9-lung macrophage axis in patients with lung cancer.

## Results

**IL-9 promotes tumor growth and alters lung macrophage populations**. The anti-tumor effect of Th9-derived IL-9 has been well characterized, particularly in subcutaneously injected tumor models[2,3,5,9]. However, contrasting studies indicated that IL-9-deficiency inhibited lung tumor metastasis[10]. To detect how IL-9 signaling affects disease progression of lung cancer patients, we performed a Kaplan-Meier survival analysis of 982 lung cancer patients from multiple studies using a public database and online tools as described[20]. High expression of $IL9$ and $IL9R$ clearly

associated with poor survival probability in lung cancer patients (Fig. 1a). Moreover, compared with normal tissues, cancer patients with lung metastasis showed increased $IL9$ and $IL9R$ expression (Fig. 1b) analyzed using published transcriptomic data obtained by gene-array as described[21]. These seemingly conflicting effects of IL-9 on tumor growth among tumor models in different organs could be due to the diverse pool of IL-9-responding cells in the specific tumor microenvironment. To define the populations that directly responded to IL-9 in the lung tumor environment, we analyzed a public scRNA-Seq dataset from a cohort of NLCSC patients[22]. Compared to non-involved normal lung tissue, monocytes/macrophages were the major immune cells infiltrating to the tumor site (Fig. 1c). TAMs were identified by the expression of $CD68$ and $CD163$ (Fig. 1d) and were likely monocyte-derived as indicated by the high expression of $CD14$ and low expression of $SIGLEC1$ (Fig. 1d). They also expressed immunosuppressive markers such as $MRC1$ (Fig. 1d). Consistent with previous reports, T cells, B cells and mast cells express $IL9R$. Strikingly, TAMs not only expressed $IL9R$ but also occupied a large proportion of $IL9R^+$ cells in the lung tumor microenvironment (Fig. 1e), indicating they could be an IL-9-responsive population in the lung tumor microenvironment. Thus, IL-9 signaling might impact the tumorigenic effects of TAMs in lung cancer patients.

To test the impact of IL-9 signaling on lung macrophages, we investigated a B16 i.v. injection model in WT, $Il9^{-/-}$ and $Il9r^{-/-}$ mice. Consistent with the lung cancer patient data, both the $Il9^{-/-}$ and $Il9r^{-/-}$ mice showed dramatically reduced tumor growth in the lung (Fig. 2a). Gene-deficient mice also demonstrated longer survival (Fig. 2b). We assessed IL-9R expression on lung cell populations[23]. IL-9R was detected across populations including basophils, mast cells, monocytes, macrophages, neutrophils and T cells (Fig. 2c). Consistent with the data from lung cancer patients (Fig. 1e), lung macrophages occupied over 75% of the IL-9R$^+$ cells (Fig. 2c).

To precisely investigate if and how IL-9 regulates distinct pulmonary macrophage populations in tumors, we gated on Mer Tyrosine Kinase (MerTK)$^+$ CD64$^+$ macrophages and distinguished AMs and IMs by SiglecF and CD11c expression. In naive mice, AMs occupied a large proportion of total lung macrophages (Supplementary Fig. 1a, b). In tumor bearing mice, three macrophage populations were present in the lung: AMs, CD11c$^-$ IMs and a third population that we termed CD11c$^+$ IMs, which will be supported in subsequent analyses (Fig. 2d). Interestingly, loss of $Il9$ or $Il9r$ resulted in increased AM, and decreased CD11c$^-$ and CD11c$^+$ IM (Fig. 2d, e, Supplementary Fig. 1c). To assess the dynamics of macrophage populations throughout tumor development, tumor growth and macrophage populations were analyzed at one, two and three weeks after tumor injection. CD11c$^-$ and CD11c$^+$ IMs significantly increased in WT mice accompanied with dramatic tumor growth from day 14 to day 21 (Supplementary Fig. 1d-f). Compared to other IL-9R-expressing cells, all three lung macrophages expressed a high level of IL-9R (Fig. 2f), further indicating they might be important IL-9 responders. To test if the regulation by IL-9 on lung macrophages is unique to B16 lung tumor models, we i.v. injected mice with Lewis lung carcinoma (LLC) tumor cells. Similar to tumor growth in the B16 i.v injection model, the $Il9r^{-/-}$ mice showed less tumor growth (Fig. 2g and Supplementary Fig. 1g). The pattern of altered macrophage dynamics was also conserved in the LLC i.v. injection model (Fig. 2h). Previous studies have shown IMs can be divided into three subpopulations based on the expression of MHCII, CD11b, CD11c and lyve1[24,25] (Supplementary Fig. 1h). In the B16 i.v. model, Lyve1 expression was similar among the three IM subsets (Supplementary Fig. 1i). There were significant decreases in the IM3 population (MHCII$^+$ CD11c$^+$ CD11b$^+$ SiglecF$^-$) when IL-9R

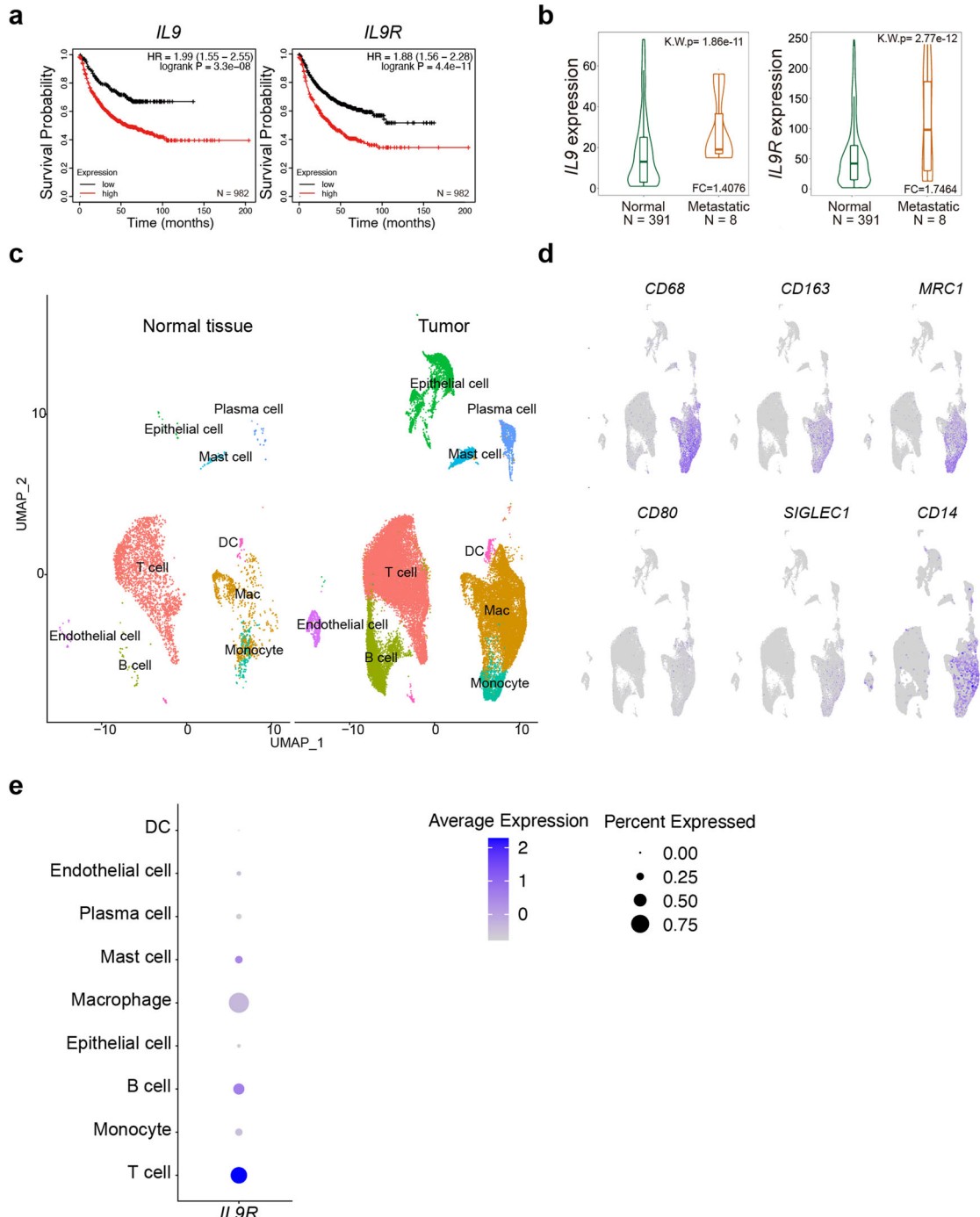

**Fig. 1 TAMs express *IL9R* in lung cancer patient tissue. a** Kaplan-Meier plots showing differences in survival among lung cancer patients ($n = 982$) using data derived from published transcriptomic data and online tools described in the Methods for *IL9* and *IL9R*. HR: hazard ratio. **b** Comparison of gene expression in normal lung tissue and metastatic lung tissues by using data derived from published transcriptomic data obtained by gene-array and online tools described in the "Methods". FC: fold change. K.W.p Kruskal–Wallis *p* value. The bars represent the proportions of metastatic tumor samples that show higher expression of the selected gene compared to normal samples at each of the quantile cutoff values (minimum, 1st quartile, median, 3rd quartile, maximum). **c** UMAP showing clusters from normal and lung tumor tissue from human lung cancer patients. DC, dendritic cell. **d** UMAP showing gene expression. **e** Dot plot showing *IL9R* expression in different clusters.

signaling was lost (Supplementary Fig. 1j). Differences from the observed populations in naive lung may be due to the tumor-promoting environment[24,25]. To further detect phenotypic changes of distinct lung macrophage populations, we stained markers related to co-stimulatory or immunosuppressive functions in the B16 injection model. The CD11c⁻ IM from WT mice exhibited lower co-stimulatory marker expression (CD86) and higher expression of CD206, which is associated with immunosuppression (Supplementary Fig. 1k, l). Other cell types were not altered in the absence of IL-9 signaling in this model (Supplementary Fig. 1m, n). By injecting B16 cells to INFER IL-9 reporter mice, we found CD4 T cells were the major IL-9 producers in the lung tumor environment (Supplementary Fig. 1o). To confirm the importance of IL-9 from T cells, mice with T cell-specific ablation

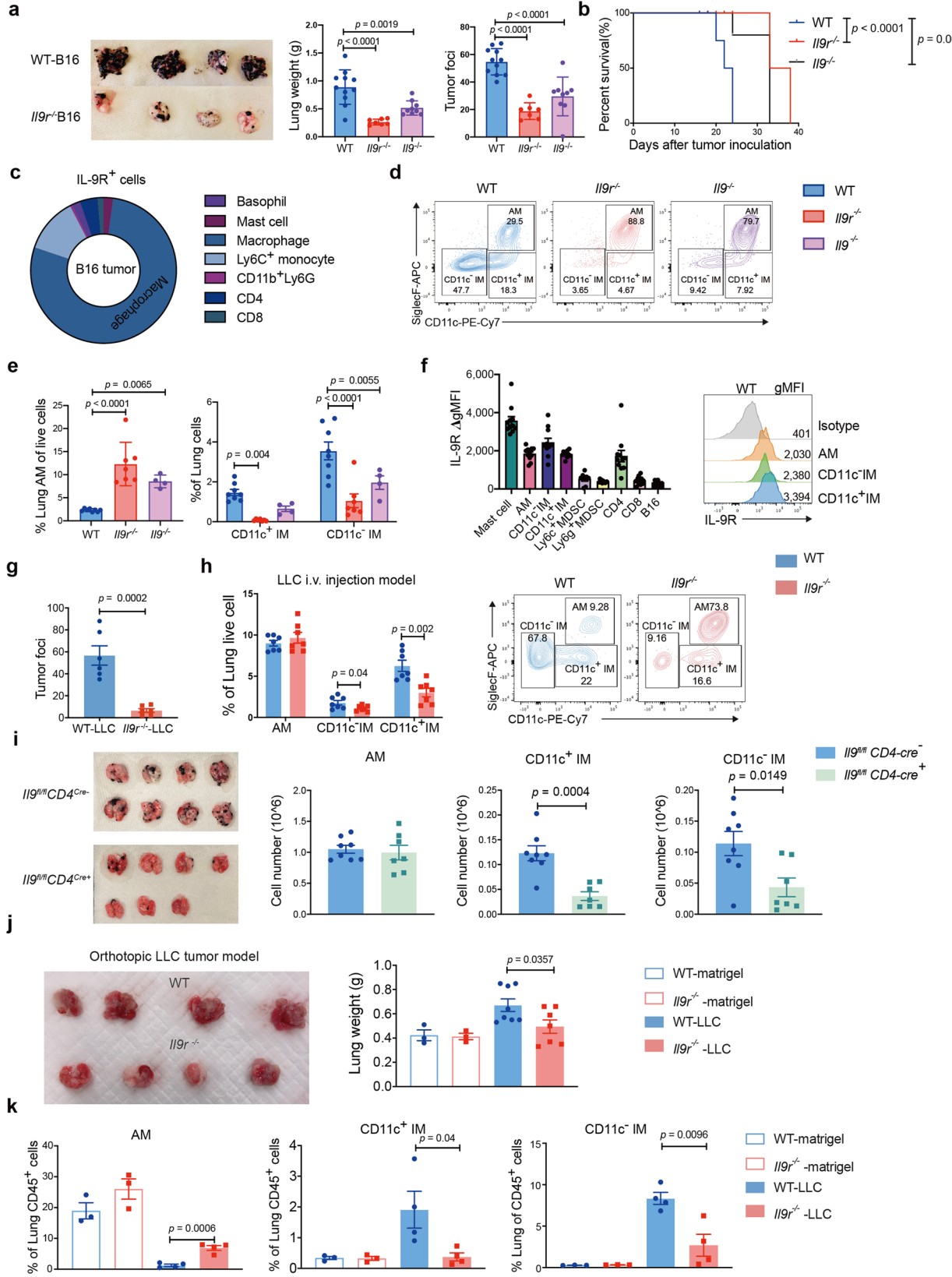

of *Il9* were injected with B16 tumor cells[26]. Compared to control mice, *Il9*<sup>fl/fl</sup> CD4-Cre<sup>+</sup> mice showed less tumor growth and decreased IM populations (Fig. 2i), similar to the pattern observed in *Il9r*<sup>−/−</sup> mice. In contrast, mice lacking mast cells, which express moderate levels of IL-9 in INFER mice (Supplementary Fig. 1o), have increased tumor growth and did not have the same changes

in macrophage populations observed when IL-9 or IL-9R are lost (Supplementary Fig. 1p, q). Together, these results suggest that CD4-derived IL-9 promotes tumor growth and IM expansion.

We next sought to evaluate if IL-9 affected tumor growth and lung macrophages in a primary lung tumor model. To test this, we used an orthotopic lung tumor model, where LLC cells were

**Fig. 2 IL-9 promotes tumor growth by altering lung macrophage populations. a–f** B16 melanoma cells were intravenously injected into the mice. **a** Tumor development was assessed on day 21 ($n = 11$ mice for WT group, $n = 7$ mice for $Il9r^{-/-}$ group, $n = 9$ mice for $Il9^{-/-}$ group). **b** Survival analysis from d0 to d40 after tumor cell injection. **c** IL-9+ cells were analyzed by flow cytometry. **d, e** Lung macrophage populations were analyzed by flow ($n = 8$ mice for WT group, $n = 7$ mice for $Il9r^{-/-}$ group, $n = 4$ mice for $Il9^{-/-}$ group). **f** ΔgMFI of IL-9R were analyzed by flow cytometry, ΔgMFI is the gMFI of each population minus the gMFI of isotype controls in that population ($n = 8$ mice for MDSC and $n = 11$ mice for other cell types). gMFI geometric mean fluorescence intensity. MDSC, Myeloid-derived suppressor cells. **g, h** Mice were i.v. injected with LLC tumor cells, tumor growth (**g**) ($n = 6$ mice) and lung macrophages (**h**) ($n = 7$ mice) were analyzed on day 20. i.v.: intravenously injection. **i** B16 melanoma cells were intravenously injected into the mice. Tumor development was assessed on day 21. Lung macrophage numbers were analyzed by FACS ($n = 8$ mice for $Il9^{fl/fl}$ CD4-Cre$^-$ group, $n = 7$ mice for $Il9^{fl/fl}$ CD4-Cre$^+$). **j, k** LLC cells were directly injected to the lung. Tumor growth was assessed 14 days after tumor inoculation (**j**) ($n = 3$ mice for WT-Matrigel and $Il9r^{-/-}$-Matrigel group, $n = 8$ mice for WT-LLC group, $n = 7$ mice for $Il9r^{-/-}$-LLC group). Lung macrophages were analyzed by flow (**k**) ($n = 3$ mice for WT-Matrigel and $Il9r^{-/-}$-Matrigel group, $n = 4$ mice for WT-LLC group and $Il9r^{-/-}$-LLC group). Data are the mean ± SEM and representative of two independent experiments. One-way ANOVA with a Dunnett's multiple comparison test was used to generate p values for multiple comparisons in a and e. Log-rank (Mantel–Cox) test was used to generate $p$ value in **b**. Unpaired two-tailed Student $t$-test was used for comparison in **g–k**. Two-way ANOVA with Sidak's multiple comparisons was used for comparisons in **e**.

suspended in matrigel and injected directly into the lung[27]. WT mice showed more lung tumor growth than the $Il9r^{-/-}$ mice (Fig. 2j). Lung macrophage populations demonstrated a similar pattern as the results from i.v. injection tumor models: increased AM and decreased CD11c+ and CD11c− IM in $Il9r^{-/-}$ mice (Fig. 2k). $Il9r^{-/-}$ mice also retain more monocytes in the bone marrow (Supplementary Fig. 1r). These data suggest that IL-9 signaling affects lung macrophage populations in multiple lung tumor growth models.

**Transcriptional signature of lung macrophages in the tumor environment.** To further compare the similarity of the gene expression among AMs, CD11c+ IMs and CD11c− IMs in a common tumor environment, we performed RNA sequencing (RNA-Seq) on flow-sorted macrophage populations from intact, perfused lungs of tumor-bearing mice. Using unsupervised clustering with differentially expressed genes indicated in a heat map, we found the CD11c+ IMs gene expression profile was more similar to CD11c− IMs, than to AMs (Fig. 3a, b). Only 691 genes were differentially expressed between CD11c− IMs and CD11c+ IMs. However, AMs showed 1495 differentially expressed genes with CD11c+ IMs and 2622 genes with CD11c− IMs (Fig. 3b, c). While many genes were common among the three populations owing to the mutual cell identity, 684 genes were common between CD11c+ IMs and CD11c− IMs, but a much smaller proportion of genes were shared between AMs and either CD11c+ IMs or CD11c− IMs (Fig. 3d). Among the genes shared between the IM populations, CD11c+ IMs expressed monocyte/interstitial macrophage-related genes, such as *Cx3cr1, Ccr2, Ly6c2*, and *Csf1r*, and a low level of AM specific genes, such as *Ear2, Wfdc21, Car4* and *Ear1* (Fig. 3e). Altogether, these data suggest that CD11c+ IMs are more closely related to CD11c− IMs and are referred to as CD11c+ IMs.

**IMs are the TAMs that respond to IL-9.** To investigate the specific macrophage populations that infiltrate the tumor site, we isolated tumor foci from the orthotopic lung tumor model. From two independent experiments, all WT mice had tumor growth, in contrast to only 20% of the $Il9r^{-/-}$ mice (Fig. 4a, b). WT mice demonstrated larger tumors than those in $Il9r^{-/-}$ mice that did develop tumors (Fig. 4a–c). IMs are the major tumor-associated macrophages (TAMs) (Fig. 4d). The gene expression profile in mouse IMs (TAMs) overlapped with 95% of genes expressed in TAMs from lung cancer patients (Fig. 4e), indicating these IM populations in mouse were conserved with human TAMs. No significant differences were found among other cell types or in cytokine expression in lung CD45+ cells (Supplementary Fig. 2a,

b). Thus, these results suggest both CD11c− IM and CD11c+ IMs are TAMs and are altered in numbers in response to IL-9.

To further define whether the effects of IL-9 were intrinsic to the macrophage population or a result of exogenous factors linked to differences in tumor growth between the strains, we generated mixed bone marrow chimeras where we injected equal numbers of WT (CD45.1) and $Il9r^{-/-}$ (CD45.2) bone marrow cells into lethally irradiated wild-type recipients (CD45.1+ CD45.2+) (Fig. 4f). Following reconstitution, chimeric mice were injected with B16 tumor cells (Fig. 4f). In a context with uniform tumor growth, we observed that the ratio of WT:$Il9r^{-/-}$ cells was higher for IM and lower for AM, consistent with earlier results (Fig. 4g, h). This result suggests that IMs develop from bone marrow derived monocytes. To further investigate this, fluorescent bead-labeled monocytes were transferred into $Il9r^{-/-}$ mice (Fig. 4i)[28]. Mice receiving WT monocytes showed increased tumor growth (Fig. 4j). Flow cytometric analysis of bead-labeled monocytes showed that IL-9 signaling promoted monocytes to become lung IMs. The majority of the CD11c−IM and about half of CD11c+ IMs are derived from circulating monocytes (Fig.4k). Both WT and $Il9r^{-/-}$ mice showed similar number of monocytes from bone marrow, blood and lung (Supplementary Fig. 2c), however, WT bone marrow monocytes demonstrated higher capability for proliferation than monocytes from $Il9r^{-/-}$ mice (Supplementary Fig. 2d). The loss of IL-9 signaling did not affect proliferation of blood and lung monocytes (Supplementary Fig. 2d). Together, these data suggest IL-9 signaling recruits circulating monocyte for development into IMs. Compared to naive mice, WT AMs showed decreased cycling as assessed by Ki67 staining in the lung tumor environment, indicating IL-9 signaling represses AM expansion by inhibiting its proliferation (Supplementary Fig. 2e). To explore if the regulatory pattern of IL-9 on macrophages is also consistent in other tumor environments, we also assessed IL-9R in a B16 s.c. model. Interestingly, TAMs in the s.c. tumor model did not express IL-9R compared to isotype control staining sample (Supplementary Fig. 2f). Moreover, the percentage of TAMs was not affected by the loss of IL-9R (Supplementary Fig. 2g), indicating the regulation of IL-9R on macrophages might be restricted to certain tissues such as the lung microenvironment.

**Lung macrophages promote IL-9-mediated lung tumor growth.** To explore whether the macrophages themselves lead to altered tumor growth between WT and $Il9r^{-/-}$ mice, or whether the crosstalk between the microenvironment and the lung macrophages were the major driver, we employed a lung macrophage adoptive transfer experiment in the tumor model. We sorted lung AMs, CD11c+ and CD11c− IMs from WT Boy/J (CD45.1+) tumor bearing mice and then transferred them to $Il9r^{-/-}$ mice

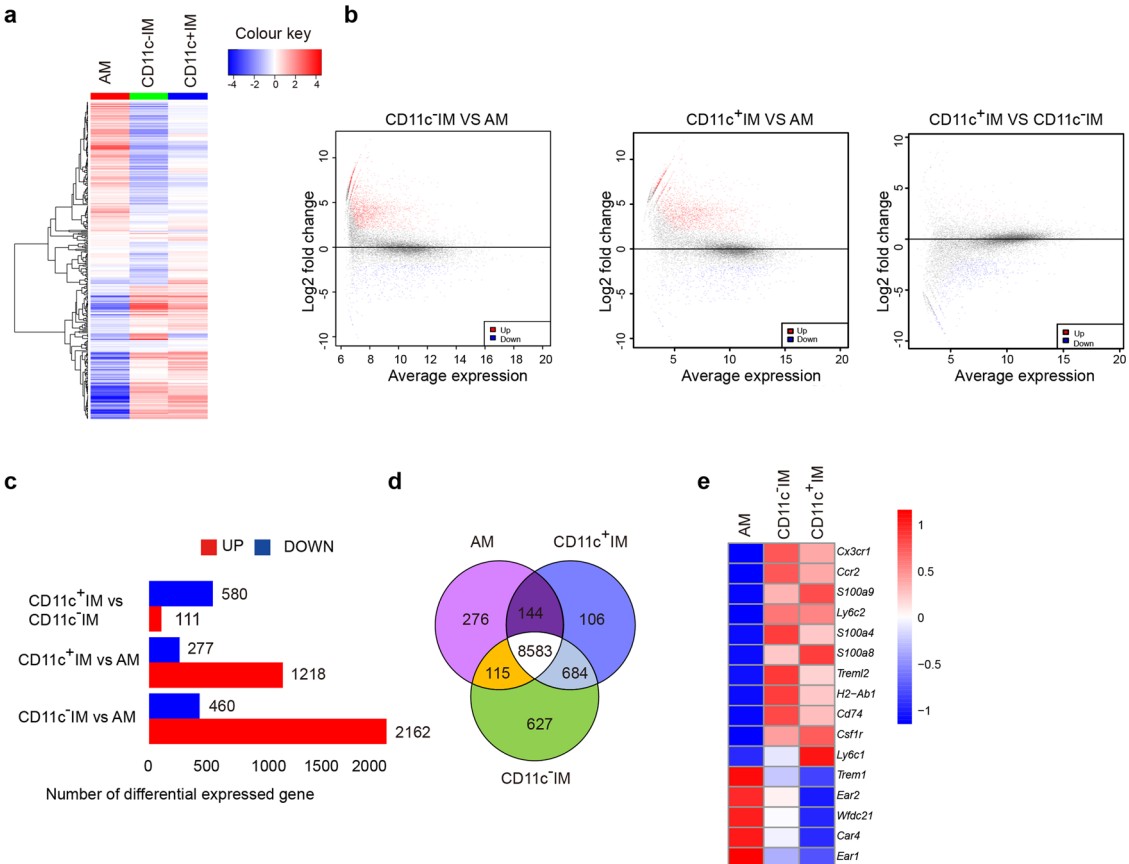

**Fig. 3 Transcriptional signature of lung macrophages in tumor environment. a–e** RNA-Seq analysis on macrophage populations isolated by FACS-sorting from entire lungs of B16 tumor bearing mice. The lungs were perfused and isolated from tumor-bearing mice. **a, b** Differential gene expression in WT AMs, CD11c⁺ IMs and CD11c⁻ IMs derived from unsupervised clustering. **c** Numbers of differentially expressed genes among WT AMs, CD11c⁺ IMs and CD11c⁻ IMs. **d** Venn diagram showing the overlap of genes expressed among AMs, CD11c⁻ IMs, and CD11c⁺ IMs. **e** Heatmap showing gene expression in macrophage populations.

(CD45.2⁺) 4 days after tumor cell injection (Fig. 5a). Donor-derived AMs and IMs were present in the lung tissue where they represented 20-40% of total lung macrophages in the recipient mice (Fig. 5b, c). WT tumor-primed lung macrophages robustly enhanced tumor growth in $Il9r^{-/-}$ mice (Fig. 5d). While it is possible AMs contribute to the IL-9-dependent tumor promoting phenotype in early tumor development, we did not observe significant changes of AM numbers comparing WT and $Il9r^{-/-}$ mice (Supplementary Fig. 1c). Thus, the IL-9-dependent changes in the IM populations prompted us to focus further functional analysis on the IM populations. To further avoid the bystander effect of other cell types on macrophages in the WT donor tumor-bearing mice, we sorted WT and $Il9r^{-/-}$ IMs from the mixed bone marrow chimeric mice described in Fig. 4f and transferred the cells to $Il9r^{-/-}$ tumor-bearing mice (Fig. 5e). Mice that received WT macrophages showed more tumor growth than mice that received $Il9r^{-/-}$ macrophages (Fig. 5f). These data suggest endogenous IL-9/IL-9R signaling is crucial for the pro-tumor function of lung macrophages.

Consistent with a published report[3], neither B16 nor LLC2 tumor cells express IL-9R or respond to IL-9, indicating IL-9 likely indirectly affects tumor growth in the lung (Fig. 2f, Supplementary Fig. 3a–c). To investigate if and how the IL-9-macrophage axis affects the growth of cancer cells in the lung, lung macrophages from tumor-bearing mice were cocultured with tumor cells for 72 h with or without IL-9 to define effects on tumor cell proliferation and apoptosis. Cells in all conditions exhibited comparable percentages of cell death and proliferation,

suggesting the IL-9-macrophage axis has no direct impact on tumor cell proliferation or cell death in these models (Supplementary Fig. 3d). Next, we performed a migration assay to test if the IL-9/macrophage axis impacts cancer cell migration. Lung macrophages were isolated from tumor bearing mice and plated in the lower chamber of the plate with PBS or IL-9. Tumor cells were placed in the upper chamber of the transwell (Supplementary Fig. 3e). While IL-9 alone was not sufficient to induce tumor migration, macrophages from both WT and $Il9r^{-/-}$ mice promoted tumor migration to a similar extent (Fig. 5g, h). Remarkably, IL-9 enhanced the ability of macrophages to induce tumor migration of both B16 tumor cells and LLC tumor cells (Fig. 5g–i), while IL-9 treatment on $Il9r^{-/-}$ macrophages had no effect (Supplementary Fig. 3f). To determine if this effect is conserved in human cells, PBMC-derived monocytes were differentiated into M1 or M2 macrophages for 7 days. IL-9 was then added to the culture for 24 h (Supplementary Fig. 3g). Human cancer cells (H838) were plated in the upper chamber with M1 or M2 macrophages in the lower chamber (Supplementary Fig. 3g). IL-9 enhanced M2 but not M1 macrophage-induced tumor cell migration (Fig. 5j and Supplementary Fig. 3h).

**IL-9 regulates the transcriptional profile of IMs.** To understand how IL-9 signaling affects the transcriptional landscape of lung macrophages, we compared the gene expression of WT and $Il9r^{-/-}$ macrophages from intact lungs of tumor-bearing mice. Unsupervised clustering with differentially expressed genes in a heat map indicated that $Il9r$-deficiency significantly affected gene

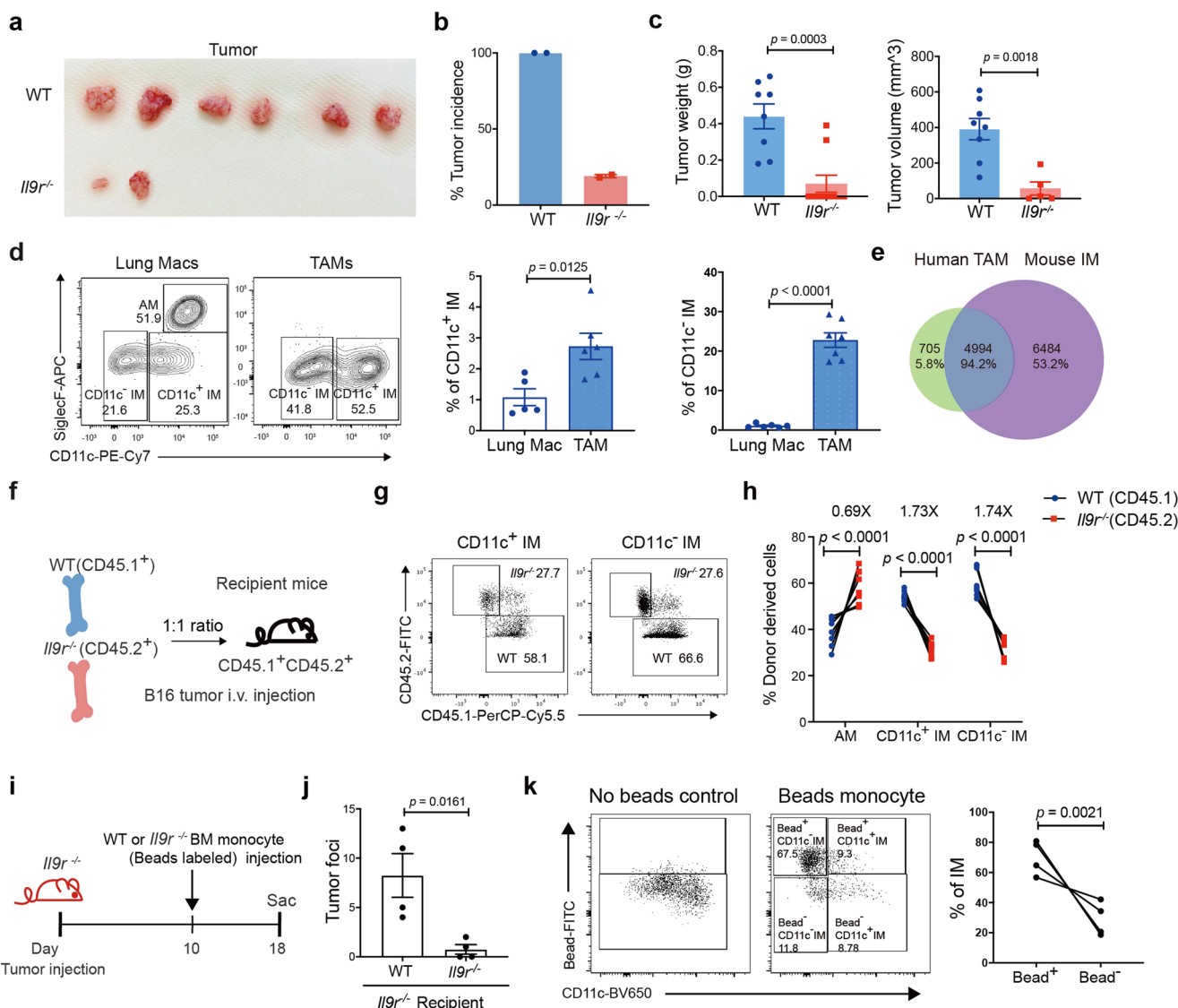

**Fig. 4 IMs are the TAMs that respond to IL-9. a–d** LLC cells were directly injected to the lung. Tumor appearance (**a**) and tumor incidence (**b**) (*n* = 2 experiments) were assessed after resection. Tumor weight (**c**) (*n* = 8 mice for WT group, *n* = 10 mice for *Il9r*$^{-/-}$ group) was calculated from two independent experiments. **d** Lung cells from tumor bearing mice and cells from tumor foci were isolated for flow analysis (*n* = 5 mice for lung Mac group, *n* = 6 mice for TAM group). TAM: tumor associated macrophage. **e** Venn graph showing the overlap genes from human TAMs and mouse IMs. **f–h** WT (CD45.1) and *Il9r*$^{-/-}$ mice (CD45.2) bone marrow cells were mixed in 1:1 ratio and transferred to lethally irradiated recipient mice (CD45.1$^+$ CD45.2$^+$ mice). After reconstitution, chimeric mice were intravenously injected with B16 tumor cells, donor lung macrophages were analyzed by flow cytometry (*n* = 9 mice). **i–k** *Il9r*$^{-/-}$ mice were injected with B16 tumor on day 0. Fluorescent bead-labeled monocytes were transferred to recipient mice on day 10 (**i**). Tumor growth (**j**) and lung macrophages (**k**) were analyzed on day 18, dot plots were gated on MerTK$^+$ CD64$^+$ SiglecF$^-$ live cells (*n* = 4 mice per group). Data are the mean ± SEM. Unpaired two-tailed Student *t*-test was used for comparison in **c**, **d** and **j**. Two-way ANOVA with Sidak's multiple comparisons was used for comparisons in **h**.

expression in CD11c$^+$ IMs and CD11c$^-$ IMs (Fig. 6a, b). Gene set enrichment analysis between WT and *Il9r*$^{-/-}$ CD11c$^+$ or CD11c$^-$ IMs identified changes in reactive oxygen species pathways and cancer module pathways (Fig. 6c, d). The differentially expressed genes between WT and *Il9r*$^{-/-}$ macrophages are involved in multiple biological pathways (Fig. 6e). We then examined gene expression that related to inflammatory or immunosuppressive functions of macrophages in tumor growth (Fig. 6f). Surprisingly, not all immunosuppressive genes were upregulated in WT cells by IL-9 (Fig. 6f), indicating that IL-9 signaling affects the pro-tumor function of macrophage not simply by promoting an immuno-suppressive phenotype in macrophages, but rather by having more subtle effects on the transcriptional profile. A set of angiogenesis-related genes were down-regulated in *Il9r*$^{-/-}$ mice (Fig. 6e, f). Greater angiogenesis in WT mice was confirmed by

immunofluorescence staining of CD31, a marker of endothelial cells (Fig. 6g). Furthermore, adoptive transfer of WT IMs rescued the loss of CD31$^+$ cells in *Il9r*$^{-/-}$ recipient mice (Fig. 6h). This indicates the IL-9/IM axis may promote angiogenesis in lung tumors. By comparing the differentially expressed genes in WT and *Il9r*-deficient IMs, we identified expression of 107 genes that were altered by IL-9 signaling in both CD11c$^+$ IMs and CD11c$^-$ IMs (Fig. 6i), indicating that there is likely a common mechanism on how IL-9 affects the function of the two interstitial macrophage populations. Among the 107 common genes were receptors (*Acva1c, Ffar2, Ffar4*), chemokines (*Cxcl1, Cxcl15*), nicotinamide nucleotide transhydrogenase, and transmembrane proteins (Fig. 6j). Among genes that were regulated by *Il9r*-deficiency in IMs, 206 are IM-enriched genes, expressed at lower levels in AMs (Fig. 6k).

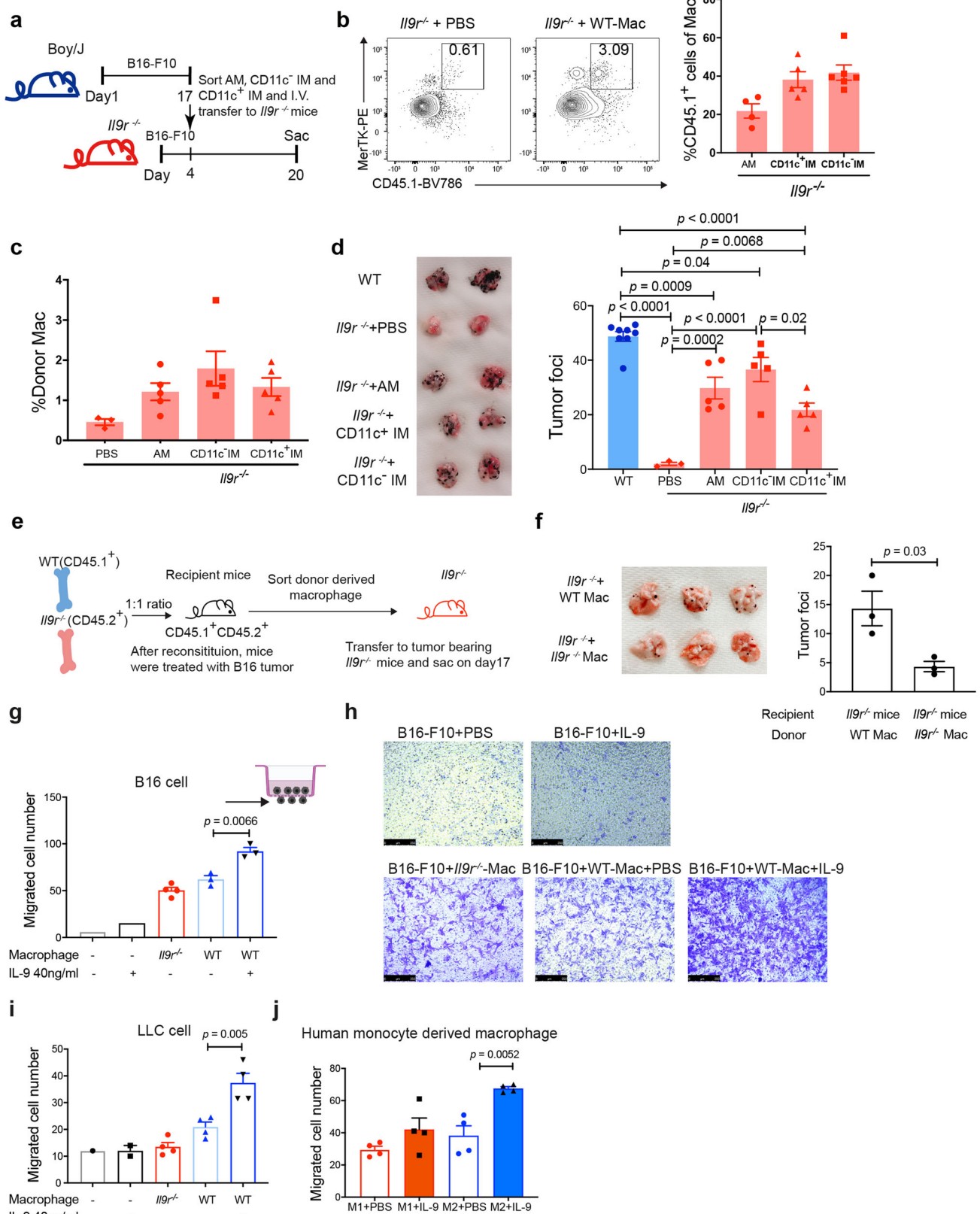

**IL-9 alters lung macrophage function by regulating Arg1 expression**. Next, we wanted to investigate the underlying mechanism of IL-9-mediated lung macrophage function in lung cancer. Among the 107 overlapping differentially expressed genes in CD11c$^+$ IMs and CD11c$^-$ IMs (Fig. 6g), *Arg1* was one of the most differentially expressed genes, and both CD11c$^+$ and CD11c$^-$ IMs showed higher Arg1 expression than AMs (Fig. 7a). Arg1 is a key factor linked to the pathogenesis of tumor growth[29,30]. Thus, we speculated that Arg1 might be a potential effector in the IL-9-macrophage axis. To test this, we first examined Arg1 protein expression in our models. There was a substantial reduction of Arg1 expression in lung IM populations

**Fig. 5 Lung macrophages promote IL-9 mediated lung tumor growth. a**, **b** Boy/J mice were injected with B16 melanoma cells. Macrophages were sorted from the entire lung of tumor bearing mice on d17 and intravenously injected to $Il9r^{-/-}$ mice 4 days after tumor injection (**a**). **b**, **c** Donor macrophages were detected on day 20 ($n = 4$ mice for AM group, $n = 5$ mice for CD11c$^+$ IM group, $n = 6$ for CD11c$^-$ IM group in panel b) ($n = 3$ mice for PBS group, $n = 5$ mice for other groups in panel c). **d** Tumor development was assessed ($n = 8$ mice for WT group, $n = 3$ mice for PBS group, $n = 5$ mice for other groups). **e**, **f** WT and $Il9r^{-/-}$ macrophages were sorted from the entire lung of mixed bone marrow chimeric mice described in Fig. 4f and transferred into $Il9r^{-/-}$ tumor bearing mice (**e**). Tumor growth was analyzed on day 17 (**f**) ($n = 3$ mice). Mac: macrophage. **g**–**j** Total lung macrophages were isolated from intact lungs of B16 (**g**, **h**) ($n = 1$ for left two groups, $n = 4$ mice for middle group, $n = 3$ mice for right two groups) or LLC (**i**) ($n = 1$ well of cell for left group, $n = 2$ wells of cell for the second left group, $n = 4$ mice for other groups) tumor-bearing mice and plated in the lower chamber of a transwell with or without IL-9; B16 or LLC cells were plated in the upper chamber. B16 cells were allowed to migrate for 16 h, and LLC cells for 3 h before counting. Unmigrated cells were removed and migrated cells were calculated from the average of two views under 20× microscopy, Scale bar = 250 μm. **j** Human monocytes were isolated from human PBMC and differentiated into M1 or M2 macrophage for 7 days. Cells were treated with IL-9 overnight, and human 838 lung cancer cells were plated in the upper chamber of the transwell. Migrated cells were counted after 3 h ($n = 4$ donors per group). Data are the mean ± SEM. One-way ANOVA with a Dunnett's multiple comparison test was used to generate $p$ values for multiple comparisons in **b**, **c**, **d**. Unpaired two-tailed Student t-test was used for comparison in **f**, **g**, **i** and **j**.

in tumor-bearing $Il9r^{-/-}$ mice (Fig. 7a–c). In mixed bone marrow chimeric mice, the increased proportion of Arg1$^+$ IMs from WT donors further confirms that intrinsic IL-9 signaling regulated expression (Fig. 7d). Moreover, IL-9 promoted Arg1 expression from IMs isolated from tumor bearing mice ex vivo (Fig. 7e). In each of the tumor models, $Il9r^{-/-}$ mice also showed significantly lower arginase activity than WT mice, indicating differential expression is related to function (Fig. 7f). iNOS, another arginine regulating enzyme was not altered in the $Il9r^{-/-}$ mice (Supplementary Fig. 4a). Together, these data suggest that Arg1 is a downstream target of IL-9 signaling in lung macrophages.

To determine if Arg1 is required for lung macrophage-mediated tumor growth, Arg1-reporter mice were injected with B16 tumor, and Arg1$^+$ or Arg1$^-$ lung total macrophages were sorted and transferred to $Il9r^{-/-}$ mice 4 days after tumor cell injection (Supplementary Fig. 4b). Recipient mice were analyzed on day 23. Strikingly, YFP$^+$ (i.e. Arg1$^+$) macrophages successfully recused tumor growth in $Il9r^{-/-}$ mice, while YFP$^-$ macrophages did not promote tumor growth (Fig. 7g and Supplementary Fig. 4c). YFP$^+$ macrophages migrated to the lung (Fig. 7h and Supplementary Fig. 4d) and restored the Arg1$^+$ macrophage population in the lung of $Il9r^{-/-}$ mice (Fig. 7i). Compared to YFP$^-$ macrophages, the purified sorted YFP$^+$ macrophages expressed greater $Il9r$ (Supplementary Fig. 4e).

To more directly test the role of Arg1 in tumor growth, $Arg1^{fl/fl}LysM\text{-}Cre^+$ or littermate control mice were injected with B16 tumor; mice lacking Arg1 expression in myeloid cells showed significantly decreased tumor growth compared to control mice on both days 14 and 21 after tumor injection (Fig. 7j). $Arg1^{fl/fl}LysM\text{-}Cre^-$ IMs, but not $Arg1^{fl/fl}LysM\text{-}Cre^+$ IMs, rescued the loss of tumor growth in $Il9r^{-/-}$ mice (Fig. 7k). Loss of Arg1 in macrophages did not impact CD206 expression in IMs (Supplementary Fig. 4f). The percentage, proliferation and cytokine production of lung T cells was not altered by Arg1 deficiency in this model (Supplementary Fig. 4g). Of note, TAMs (IMs) were the major population of Arg1 producers in both this model and lung cancer patients (Fig. 7l, m). These results demonstrated that IL-9 regulation of lung tumor growth requires Arg1 in lung macrophages.

To more thoroughly investigate if macrophages impact the adaptive immune response in the tumor model, T cell activation markers were analyzed. WT mice showed more naive-like CD8 in both LLC2 and B16 i.v. models (Supplementary Fig. 4h-i). Adoptive transfer of WT IMs to $Il9r^{-/-}$ mice resulted in greater proportions of resting or naive phenotype CD8$^+$ cells in the lung (Supplementary Fig. 4i). Together, these data suggest IL-9/IM/Arg1 axis may diminish the active anti-tumor immune response.

$IL6$ is one of the commonly expressed genes by both human and mouse TAMs (Fig. 4e), and macrophages are the major $IL6$

expressing population in lung cancer patients (Fig. 8a). The mouse RNA-Seq data set demonstrated that $Il6$ was down-regulated in $Il9r^{-/-}$ IMs (Fig. 6f). To further demonstrate IL-9-dependent IL-6 expression in IMs, naive mice were injected with IL-9 for three days. We observed that IL-9 induced IL-6 expression from IMs (Fig. 8b). IL-9 also promoted $IL6$ expression in human bone marrow derived M2 macrophages (Fig. 8c). IL-9R-deficient tumor-bearing mice exhibited lower serum IL-6 level than WT mice (Fig. 8d). These results indicate IL-9 induced IL-6 expression from IMs in lung cancer. Interestingly, we found IMs that lacked Arg1 expression showed less IL-6 expression (Fig. 8e). $Il9r^{-/-}$ recipients of Arg1$^+$ cells had increased serum IL-6 concentrations (Fig. 8f). Arg1$^+$ macrophages from tumor bearing mice secreted more IL-6 than Arg1$^-$ macrophages, and were capable of inducing tumor cell migration in the presence of IL-9 (Fig. 8g, h). In mixed bone marrow chimeric tumor-bearing mice, Arg1-expressing cells also co-expressed IL-6 (Fig. 8i, j) and the Arg1$^+$ IL-6$^+$ donors were mainly derived from WT donors (Fig. 8i, j). Altogether, these data suggest that endogenous IL-9 signaling induces an Arg1-IL-6 double positive IM population. To test the impact of IL-6 in B16 lung tumor growth, anti-IL-6 blocking antibody was given to tumor-bearing mice. Blockade either early (d7) or late (d14) in tumor development successfully attenuated lung tumor growth (Fig. 8k–m, Supplementary Fig. 5a, b). To test if IL-6 has a direct effect on tumor cells, B16 cells were stimulated with IL-6, IL-9 or IGF-1, but none of the factors activated pSTAT3 in tumor cells above background levels (Supplementary Fig. 5c). Previous studies have shown IL-6 affects DC differentiation and activation[31]. In the B16 model, the number of lung CD11b$^+$ DCs and CD103$^+$ DCs was not affected by the loss of IL-9 signaling (Supplementary Fig. 5d). However, CD11b$^+$ DCs and CD103$^+$ DCs from $Il9r^{-/-}$ tumor bearing mice demonstrated decreased pSTAT3 and increased expression of several markers that associated with anti-tumor immunity (Supplementary Fig. 5e, f). When IL-6 was blocked, CD103$^+$ DCs showed increased expression of MHCII (Supplementary Fig. 5g), suggesting IL-6 affects tumor growth by indirectly suppressing the antigen presenting capability of lung DCs. Collectively, these data suggested that IL-9 signaling promotes pro-tumorigenic effects by increasing Arg1 and IL-6 expression.

**Therapeutic targeting of the IL-9-macrophage axis prevents lung cancer growth.** Since we have shown IL-9 promotes tumor growth by upregulation of Arg1 expression in mouse models, we next wanted to investigate whether this paradigm was conserved in the development of human lung cancer. We performed a Kaplan-Meier survival analysis of lung cancer patients from the database and online tools described in Fig. 1a, b[20]. Consistent

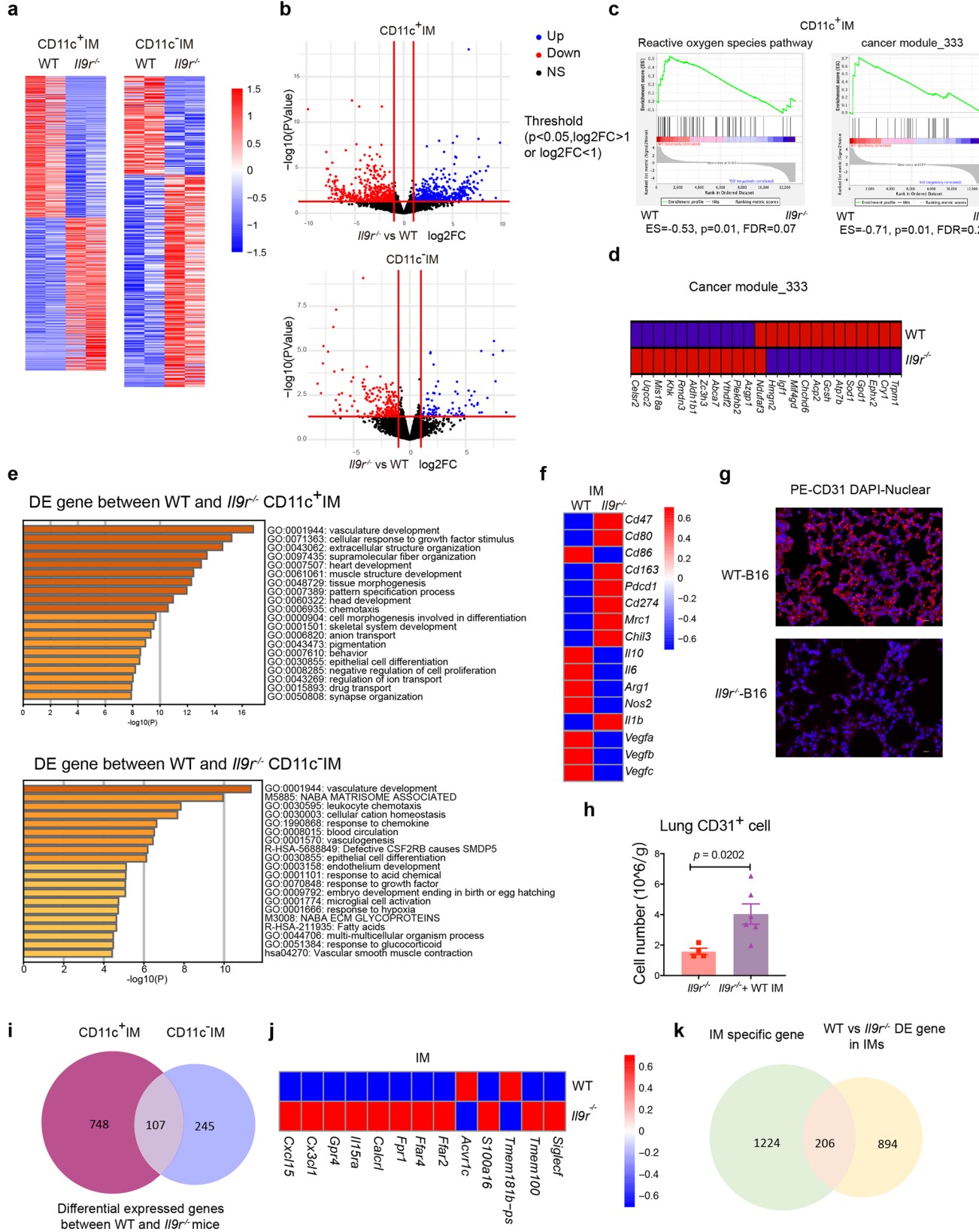

with our results in mouse models, high expression of *ARG1* and *IL6* associated with poor survival probability in lung cancer patients (Fig. 9a). Compared with normal tissues, lung metastatic tissues showed increased *ARG1* expression (Fig. 9b)[21]. We further validated these findings by detecting mRNA in cells from lung tumor sites and normal lung tissues. Consistent with publicly

available data, we found increased *IL9R* expression in lung tumor tissue compared to normal lung tissue (Fig. 9c). *IL6* expression was also increased in lung tumor tissue compared to normal lung tissue (Fig. 9c). Lung cancer patients also showed increased concentrations of serum IL-9, IL-6, and enhanced arginase activity (Fig. 9d). Compared to macrophages in normal lung

**Fig. 6 IL-9 regulates the transcriptional profile of IMs. a–f** RNA-Seq analysis on macrophage populations isolated by flow-sorting from intact lungs of B16 tumor-bearing mice. **a, b** Differentially expressed gene between WT and *Il9r*−/− macrophages. **c** GSEA analysis on the differentially expressed gene between WT and *Il9r*−/− CD11c+ IMs. **d** Heatmap showing gene expression related to cancer module 333. **e** Pathway analysis on differentially expressed genes in WT and *Il9r*−/− IMs. **f** Heatmap showing gene expression in WT and *Il9r*−/− macrophage populations. **g** Immunofluorescence analysis of CD31+ cells of lung sections from tumor bearing mice, Scale bar = 10 μm. **h** *Il9r*−/− mice were injected with B16 tumor cells on day 0, PBS/WT IMs were transferred to recipient mice on day 14. Lung CD31+ cells were analyzed by flow on day 21 (n = 4 mice for *Il9r*−/− group, n = 6 mice for *Il9r*−/− + CD11c+ IM group). **i** Venn diagram showing the overlap of differentially expressed genes in CD11c+ IMs and CD11c− IMs. **j** Heatmap showing the expression of selected genes common between the IM populations. **k** Venn diagram showing common genes between IM specific genes and differentially expressed (DE) genes regulated by *Il9r*-deficiency in IMs. Data are the mean ± SEM. Unpaired two-tailed Student *t*-test was used for comparison in **h**.

tissue, TAMs from lung tumors expressed higher level of *IL9R, ARG1* and *IL6* (Fig. 9e). Multiplex-immunohistochemistry was performed to confirm that TAMs in lung cancer patients express IL-9R at the protein level (Fig. 9f, g). Compared with macrophages within normal lung tissue, macrophages located in the tumor site expressed dramatically more IL-9R (Fig. 9f, g, Supplementary Fig. 6a). Together, these results strongly suggest the IL-9-macrophage-Arg1 axis is correlated with the development of human lung cancer.

Based on these findings, we further explored the therapeutic efficacy of targeting IL-9R signaling in lung macrophages for cancer therapy. To test this, we specifically inhibited IL-9R signaling in lung macrophages using nanoparticles conjugated with peptide that recognize SIRPα that is expressed on the surface of macrophages, for delivery of *Il9r*-siRNA based on an approach parallel to that described[32,33]. The siRNA was tagged with Alexa Fluor 555, which allowed us to detect the cells targeted with the siRNA-nanoparticles (Fig. 9h). Seven days after B16 melanoma injection, mice were treated with nanoparticles containing either Scrambled (Scr)-siRNA or *Il9r*-siRNA every 72hs (Fig. 9h). Among all CD45+ immune cells in the lung targeted by the siRNA-nanoparticles, over 60% were macrophages/monocytes (Supplementary Fig. 6b) and 60%-80% of all lung macrophages received siRNA-nanoparticles (Supplementary Fig. 6c). Macrophages that received the *Il9r*-siRNA nanoparticles showed substantial reduction of IL-9R expression compared to cells receiving the Scr-siRNA nanoparticles, indicating successful inhibition of *Il9r* expression in vivo and demonstrated significantly reduced tumor growth on day 21 (Fig. 9i, j and Supplementary Fig. 6d-e). CD11c− IM percentages were inhibited by *Il9r*-siRNA nanoparticle injection (Supplementary Fig. 6f). Arg1 and *Il6* expression were also inhibited by the knock-down of *Il9r* in vivo (Fig. 9k, l). Altogether, these results indicate that IL-9 signaling in macrophage could be a promising therapeutic target for lung cancer and potentially in other inflammatory lung diseases.

## Discussion

The biological functions of IL-9 are still not well defined and are likely to be context dependent. In tumor immunity, IL-9 has garnered interest as a potent effector in adoptive cell therapies. However, growing evidence suggests that IL-9 can promote tumor growth in some contexts. These disparate effects may depend upon the IL-9-responsive cell types, an area that has still not been studied extensively. In this report, we provide data to support a model where IL-9 promotes tumor growth in the lung through macrophages. In this model, IL-9 production, predominantly from CD4 T cells, promotes the expansion of CD11c− and CD11c+ interstitial macrophages. IL-9 also changes expression of genes in these populations, including the Arg1 gene that is critical for the pro-tumor activity. We speculate that Arg1 acts intrinsically in the macrophages, because there is a correlation between macrophage Arg1 and IL-6 production in macrophages and because we do not see suppression of T cell effector molecules in

the lung. Importantly, tumor growth is mediated when interstitial macrophages are the only IL-9-responsive cell in the system, supporting a central role of these cells in the tumor micro-environment. This model suggests that targeting IL-9 signaling could be a potential therapeutic strategy for tumors with IL-9-responsive macrophages.

The contribution of IL-9 or IL-9-producing cells in tumor immunity is model-dependent. Th9 cells and other IL-9 producers can have anti-tumor effects through both IL-9-dependent-and -independent mechanisms[4–6,8]. In contrast, *Il9*-deficient mice show dramatically less tumor burden in a tumor metastasis model[10]. A recent study found Th9 cells induce metastatic spreading by affecting lung cancer cell epithelial-mesenchymal-transition[11]. The fact that 10% of metastatic lung tumor patients from data in The Cancer Genome Atlas database showed amplification of *IL9R* also suggests IL-9 signaling might have a pro-tumorigenic role. In line with those observations, our study found that the loss of *Il9r* or *Il9* limited tumor growth in the lung.

IL-9 activates intracellular signaling by binding to IL-9R. IL-9R expression has been found in multiple cell types, including T cells, B cells, mast cells, ILC2s, goblet cells and epithelial cells[1,34–40]. Those populations can respond to IL-9 in various disease contexts. However, the identity of IL-9 responsive cells in lung cancer is unclear. By assessing IL-9R expression across various cell types, we found lung macrophages are the major component of IL-9R responsive cells. Previous studies have shown IL-9 affects microglia in multiple sclerosis, and IL-9R expression has been detected in human blood monocytes and alveolar macrophages[18,19,41]. A recent study found that IL-9-injection activated M1 macrophages and inhibited lung metastasis[42]. The seemingly conflicting effect from that report could be due to gating of differing populations of macrophages, the ectopic effects of IL-9 in the tumor microenvironment, or a lack of context if a specific cell location is required for optimal IL-9 delivery. Overall, if and how IL-9 regulates interstitial macrophages in diseases states has not been studied. Our results suggest IMs are the major IL-9 responsive cells in the context of lung cancer. They are also a key driver of IL-9 mediated tumor growth. This provides an important perspective on understanding how IL-9 promotes lung tumor growth. Casanova-Acebes et al demonstrated that AMs accumulated close to tumor cells early during tumor formation and protect them from adaptive immunity in NSCLC[43]. Consistent with those observations, we also found that AMs promote tumor growth. Both our study and Casanova-Acebes et al found IMs become the dominant TAMs in later stages of tumor development. Arg1 was a key gene that distinguished monocyte-derived macrophages (interstitial phenotype) from tissue-resident (alveolar phenotype) macrophages in both studies. As we demonstrated here, Arg1 is a key effector produced by macro-phage to promote tumor growth. Compared to IMs, Arg1 expression in AMs is much lower, which is at least one mechanism through which IMs contribute to late-stage tumor growth. Interestingly, the TAMs in the s.c.B16 tumor do not express IL-9R, which suggests the ability of IL-9 to regulate

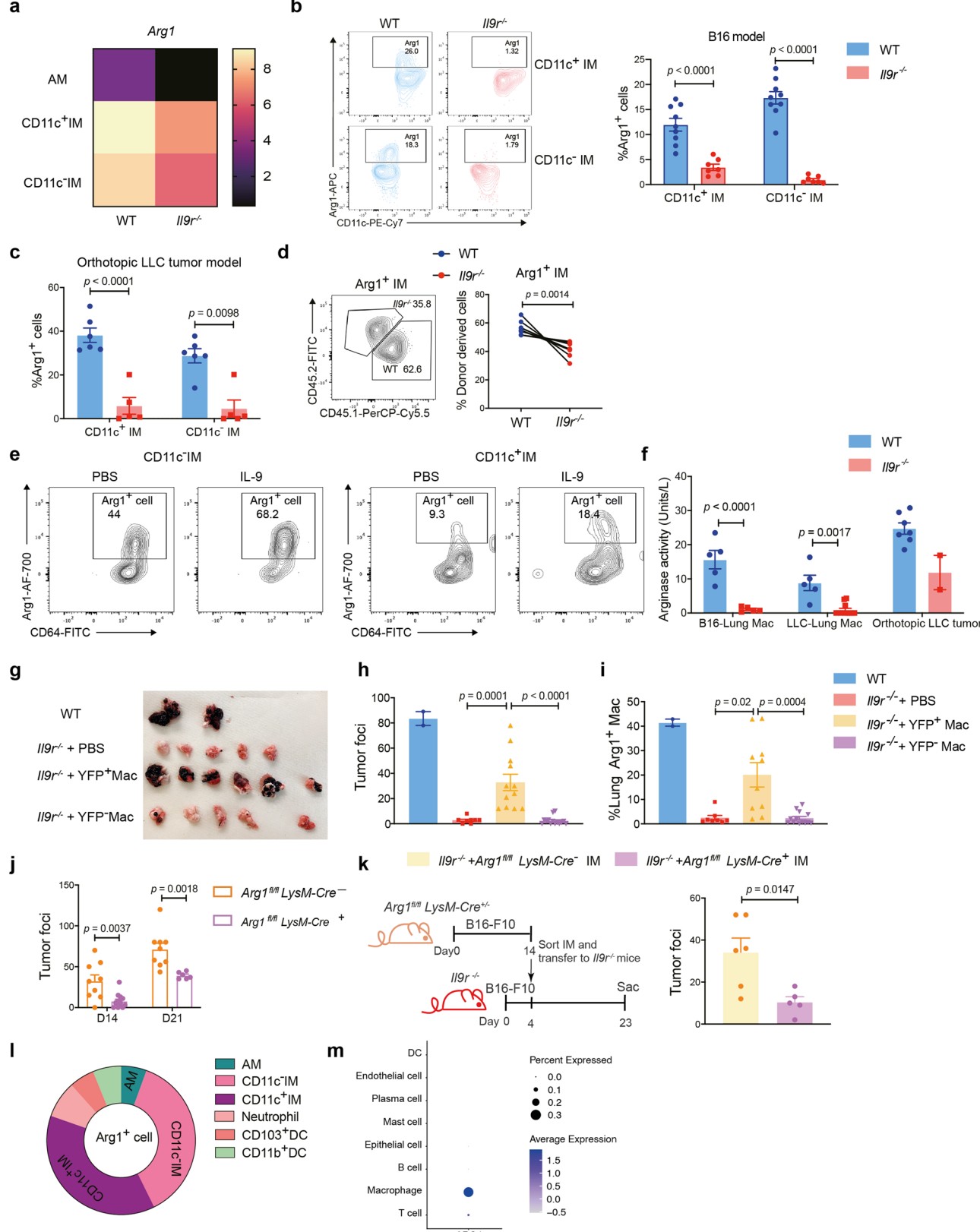

macrophages could be organ-specific or dependent on the local microenvironment. Further studies are required to test if IL-9 regulates macrophage phenotype and function in other organs.

Recent reports showed the presence of the CD11c+ IM in OVA- and HDM-induced allergic inflammation[44,45]. Here we found that the induction of CD11c+ IM is not limited to a Type 2 immune response, but also arose in the lung tumor environment where a qualitatively distinct immune response occurs. The CD11c+ IM is also the major macrophage subtype infiltrating the tumor site. The expansion of the CD11c+ IM population is highly

**Fig. 7 IL-9 impacts lung macrophage function by regulating Arg1 expression. a** Heatmap showing *Arg1* expression from RNA-Seq experiment described in Fig. 5. **b, c** Arg1 expression in macrophages from B16 tumor model (**b**) ($n = 9$ mice for WT group, $n = 7$ mice for $Il9r^{-/-}$ group) and orthotopic LLC lung tumor model (**c**) ($n = 6$ mice for WT group, $n = 5$ mice for $Il9r^{-/-}$ group). **d** Arg1 expression in mixed bone marrow chimeric mice described in Fig. 4f ($n = 9$ mice). **e** CD11c⁻IM and CD11c⁺IM were isolated from WT tumor bearing mice and stimulated with PBS or 40 ng/ml IL-9 for 48 h. Arg1 production was analyzed by flow. **f** Arginase activity were analyzed in total lung macrophages and tumor foci ($n = 5$ mice for WT and $Il9r^{-/-}$ - B16-Lung Mac group, $n = 5$ mice for WT-LLC group, n = 12 mice for $Il9r^{-/-}$ -LLC−Lung Mac, $n = 7$ mice for WT- orthotopic LLC tumor group, $n = 2$ mice for $Il9r^{-/-}$- orthotopic LLC tumor group). **g–i** YARG mice were injected with B16 cells, and total lung Arg1⁺/⁻ macrophages were sorted from intact lungs of tumor bearing mice on day 14. Cells were intravenously injected into $Il9r^{-/-}$ mice 4 days after tumor injection. **f, g** Tumor development was analyzed on day 23. **h** Donor macrophages were analyzed by flow cytometry ($n = 2$ mice for WT group, $n = 8$ mice for $Il9r^{-/-}$ + PBS group, $n = 12$ mice for $Il9r^{-/-}$ + YFP⁺ Mac group, $n = 14$ mice for $Il9r^{-/-}$ +YFP⁻ Mac group). **i** Lung Arg1⁺ macrophages were analyzed by flow cytometry ($n = 2$ mice for WT group, $n = 8$ mice for $Il9r^{-/-}$ + PBS group, $n = 10$ mice for $Il9r^{-/-}$ + YFP⁺ Mac group, $n = 13$ mice for $Il9r^{-/-}$ +YFP⁻ Mac group). Mac: macrophage. **j** $Arg1^{fl/fl}$ $LysM$-$Cre^{+/-}$ mice were injected with B16 tumor, tumor development was analyzed on day 14 and day 21 ($n = 13$ mice for D14 $Arg1^{fl/fl}$ $LysM$-$Cre^{+}$ group, $n = 6$ mice for D21 $Arg1^{fl/fl}$ $LysM$-$Cre^{+}$ group, $n = 9$ mice for D14 and D21 $Arg1^{fl/fl}$ $LysM$-$Cre^{-}$ groups). **k** IMs from tumor-bearing $Arg1^{fl/fl}$ $LysM$-$Cre^{+}$ mice or littermate control mice were sorted from intact lungs of tumor bearing mice 14 days after tumor injection and transferred to $Il9r^{-/-}$ mice which have been injected with tumor 4 days before. Tumor development was analyzed on day 17 ($n = 6$ mice for $Arg1^{fl/fl}$ $LysM$-$Cre^{-}$ group, $n = 5$ mice for $Arg1^{fl/fl}$ $LysM$-$Cre^{+}$ group). **l** Arg1⁺ cells were analyzed from WT tumor bearing mice by flow cytometry ($n = 8$ mice per group). **m** Dot plot showing *ARG1* expression in different clusters from human lung cancer patient scRNA-Seq. Data are the mean ± SEM. Unpaired two-tailed Student *t*-test was used for comparison in **b**, **c**, **f** and **k**. Two-tailed paired t test was used for generating *p* value in **d**. One-way ANOVA with a Dunnett's multiple comparison test was used for multiple comparisons in **h** and **i**. Two-way ANOVA with Sidak's multiple comparisons was used for comparisons in **j**.

---

dependent on IL-9 signaling. This population acts as a key component for IL-9 mediated diseases, demonstrated by the restoration of disease progression in adoptive transfer experiments. Although this study has advanced the understanding of the functions of lung IM in various disease environments, further studies are required to investigate the transition from monocytes to IM populations in various microenvironments, their biological functions in specific inflammatory responses, and their transcriptional programs.

How orthologous macrophage populations are between humans and mice in lung cancer is still an area of investigation. While a recent report suggested that macrophages populations are the least homologous among cell populations in lung tumors[46], another report found clear parallels between human and mouse macrophages in non-small cell lung cancer[43]. As already noted, there are clear transcriptional similarities between the macrophage populations in our study and those identified in Casanova-Acebes et al.[43]. Moreover, we identified genes of interest including *IL9R* and *ARG1* in datasets from patient samples. Differences might also arise from patient heterogeneity and the mouse models being used. The specialization of macrophages in lung cancer clearly needs further exploration, and this will also likely lead to a better understanding of species differences as well.

By comparing the transcription profiles from WT and $Il9r^{-/-}$ macrophages, we found that IL-9 signaling significantly affects the expression of a large number of genes. Among them, *Arg1* was downregulated in all three macrophage populations. Increased expression of Arg1 has been found in multiple cancers[30]. Although several cell types can produce Arg1, such as macrophages, monocytes and neutrophils, we found that in the lung tumor environment IMs are the major Arg1 producers. Moreover, we demonstrated IL-9 is an important Arg1 stimulus both in vivo and in vitro. The fact that Arg1-expressing macrophages can rescue tumor growth in $Il9r^{-/-}$ mice indicates Arg1 is a critical effector in IL-9-macrophage mediated tumor growth. Previous reports found arginine promotes anti-tumor effects of T cells by suppressing T cell function[47,48]. Most of studies demonstrated this effect by adding arginine to in vitro cultured T cells. However, in this study we found the deficiency of Arg1 in macrophages did not affect T cells cytokine production in the lung microenvironment of B16 tumor-bearing mice. This could be due to the complexity of the in vivo environment, with Arg1 produced by other cells compensating for the loss of Arg1 in

macrophages. Additional studies are needed to determine whether the source of Arg1 has differential impacts on T cell function. Here we found IL-9 stimulated Arg1⁺ macrophages secreted IL-6, which is a factor that can drive macrophages to acquire an immunosuppressive phenotype and further promote lung inflammation[49,50]. This is a potential explanation for why loss of IL-9 signaling hinders the pro-tumor effects of lung macrophages. IL-9 may induce additional pro-angiogenic factors as well. Thus, blocking IL-9 signaling on lung macrophages, when macrophages are IL-9R⁺, could be a feasible therapeutic strategy for lung cancer and potentially other lung diseases. Whether the IL-9/macrophage/Arg1 circuit that controls the lung inflammatory environment also impacts inflammation in other tissues is unclear. As IL-9 functions more broadly at mucosal surfaces, there are additional diseases where targeting IL-9 to manipulate macrophage function might be an attractive therapy.

## Methods

**Mice**. All mice were on C57BL/6 background. Wild type mice (C57BL/6, 002014), Boy/J mice (C57BL/6, 002014), Kit $^{W$-$sh}$ mice (C57BL/6, 030764) and YARG mice (C57BL/6, 015857)[51] were purchased from The Jackson Laboratory. $Il9r^{-/-}$ mice (C57BL/6) were a gift from Dr. Jean-Christophe Renauld[52]. $Il9^{-/-}$ mice (C57BL/6) were provided by Dr. Andrew McKenzie[53], through Dr. Alexander Kirsch and Dr. Sophie Paczesny. INFER mice were provided by Dr. Paula Licona-Limón and Dr. Richard A. Flavell[54]. $Il9^{fl/fl}$ crossed to CD4-Cre transgenics were recently described[26]. Both female and male mice were used between the age of 8 weeks to 16 weeks. All the mice were maintained in SPF animal facilities (ambient temperature 70–72 °F, humidity 50%, light/dark cycle 12/12 h). All experiments were performed with the approval of the Indiana University Institutional Animal Care and Use Committee.

**Patient samples**. Patient sample collection and analysis were approved by the Institutional Review Board of Indiana University and donors for biobank submission provided written consent. Lung cancer serum was purchased from the Indiana Clinical and Translational Sciences Institute. Lung sections and RNA samples were purchased from OriGene. Detailed information is listed in Supplementary Tables 1–3.

**Cell lines**. B16-F10 melanoma cell line was a gift from Dr. Dario Vignali and Dr. Kai Yang. LLC (LL/2-Luc2) and H838 lung cancer cells were purchased from ATCC. B16 cells and LLC cells were cultured in DMEM media containing 10% Fetal bovine serum (FBS, Atlanta Biologicals), 1% antibiotics (penicillin and streptomycin/stock; Pen 5000 µg/ml, Strep 5000 µg/ml), 1 mM sodium pyruvate, 1 mM L-Glutamine, 2.5 ml of non-essential amino acids (Stock; 100 X), 5 mM HEPES (all from LONZA) and 57.2 µM 2-Mercapoethanol (Sigma-Aldrich). H838 cells were cultured in RPMI1640 media containing 10% Fetal bovine serum (FBS, Atlanta Biologicals), 1% antibiotics (penicillin and streptomycin / stock; Pen 5000 µg/ml, Strep 5000 µg/ml), 1 mM sodium pyruvate, 1 mM L-Glutamine, 2.5 ml

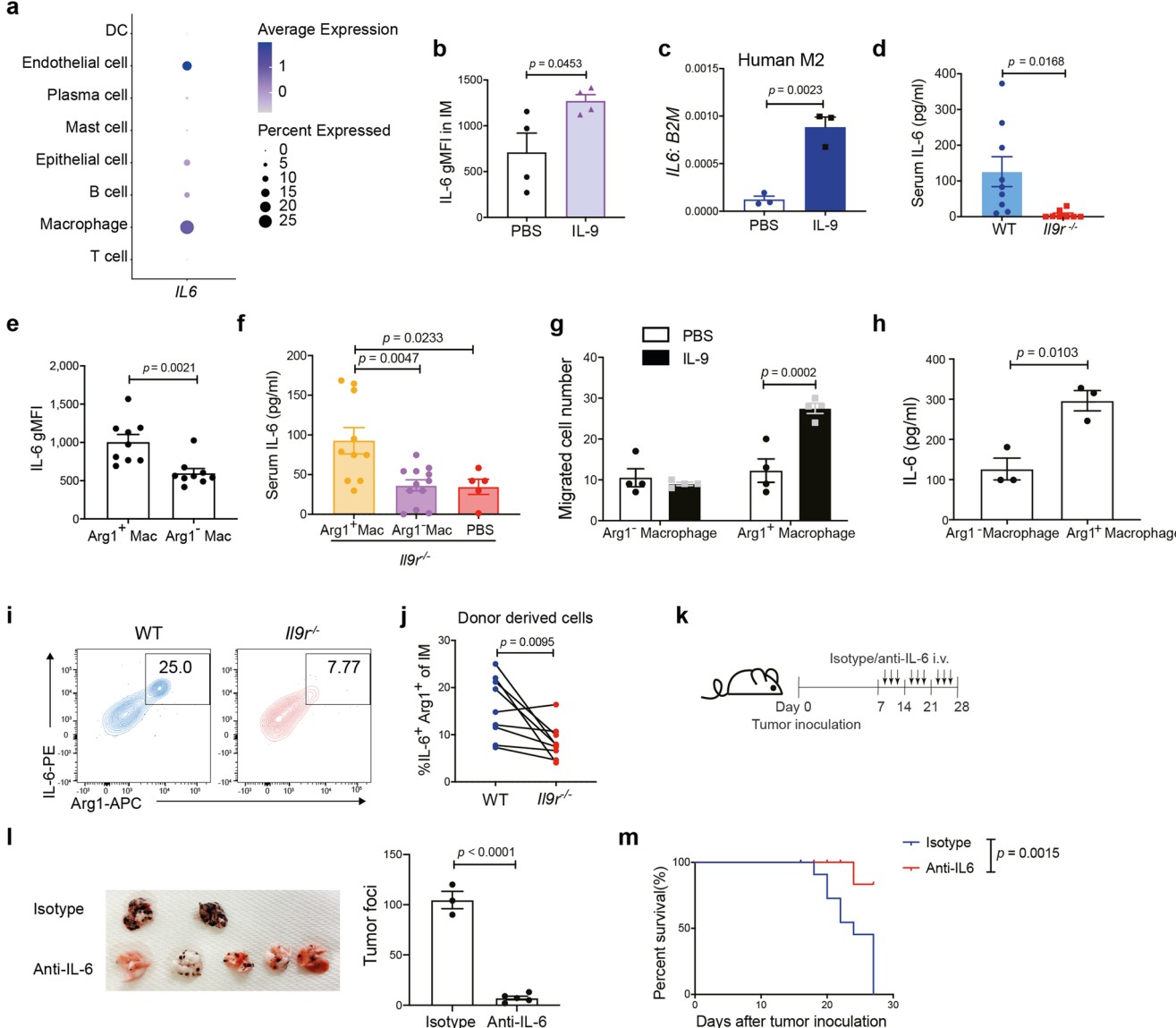

**Fig. 8 IL-9 induces IL-6 expression in Arg1 expressing IMs. a** Dot plot showing *IL6* expression in different clusters from human lung cancer patients scRNA-Seq. **b** Naive mice were i.v injected with IL-9 for 4 days, IL-6 expression in IMs was analyzed ($n = 4$ mice). **c** *IL6* expression in human PBMC monocyte derived M2 macrophage ($n = 3$ donors). **d** Serum IL-6 level in B16 tumor bearing mice ($n = 9$ mice for WT group, $n = 8$ mice for *Il9r$^{-/-}$* mice). **e** IL-6 expression was analyzed by gating on Arg1$^+$ or Arg1$^-$ macrophages from WT B16 tumor bearing mice ($n = 9$ mice). **f** Serum IL-6 level from mice described in Fig. 6f were analyzed by ELISA ($n = 10$ mice for Arg1$^+$ Mac group, $n = 12$ mice for Arg1$^-$ Mac group, $n = 5$ mice for PBS group). **g** Arg1$^+$ or Arg1$^-$ macrophages were sorted from entire lung of tumor bearing mice and plated in the lower chamber of the transwell; B16 tumor cells were placed in the upper chamber. Cells were allowed to migrate for 12 h ($n = 4$ mice). **h** Arg1$^+$ or Arg1$^-$ macrophages were sorted from entire lung of tumor bearing mice and cultured for 24 h, IL-6 concentration was analyzed ($n = 3$ mice). **i, j** Mixed bone marrow chimeric mice were generated and injected with B16 tumor as described. Donor derived IL-6$^+$ Arg1$^+$ IMs were analyzed by flow cytometry, dot plots were gated on live IMs ($n = 9$ mice). **k–m** WT tumor-bearing mice were treated with isotype antibody or anti-IL-6 antibody as shown in k, tumor growth (**l**) ($n = 3$ mice for isotype group, $n = 5$ mice for anti-IL-6 group) and survival (**m**) ($n = 12$ mice) were analyzed. Unpaired two-tailed Student *t*-test was used for comparison in **b**, **c**, **d**, **h** and **l**. Data are the mean ± SEM. One-way ANOVA with a Dunnett's multiple comparison test was used for multiple comparisons in **f**. Paired two-tailed Student *t*-test was used for comparison in **j**. Two-way ANOVA with Sidak's multiple comparisons was used for comparisons in **g**.

of non-essential amino acids (Stock; 100 X), 5 mM HEPES (all from LONZA) and 57.2 µM 2-Mercapoethanol (Sigma-Aldrich).

**Macrophage induced tumor migration assay**. The tumor cell line was starved in FBS-free media for 24 h. Total macrophage was isolated from d20 tumor bearing mice and isolated by using anti-Mertk-biotin antibody (Miltenyi Biotec). $0.2 \times 10^6$ macrophage was placed in the lower chamber in complement DMEM media with or without IL-9 (40 ng/ml, Biolegend, 556004). Tumor cells ($0.1 \times 10^6$) were placed in the upper chamber in Serum free media. Cells were allowed to migrate for 16 h for B16 cells or 3 h for LLC cells. Migrated cells were fixed with 4% formaldehyde

for 10 min, permeabilized with menthol for 1 min, stained with Crystal violet dye for 15 min, and quantified by microscopy.

**Tumor growth model**. Mice were intravenously injected with 0.1 million B16 cells or 0.3 million LLCs in 300 µl PBS. Lung tumor growth was analyzed 21 days after tumor inoculation; mice were euthanized early if weight loss was more than 20% of body weight or if they were visibly distressed. This limit was never exceeded in our studies. Mice were euthanized by $CO_2$ asphyxiation and lungs were washed with cold PBS for two times. BALF cells were counted and centrifuged at 1500 g for 5 min at 4 °C for further surface staining. Lung tumor loci were counted and lung was weighed. A portion of lung tissue was taken for histology analysis. The rest of

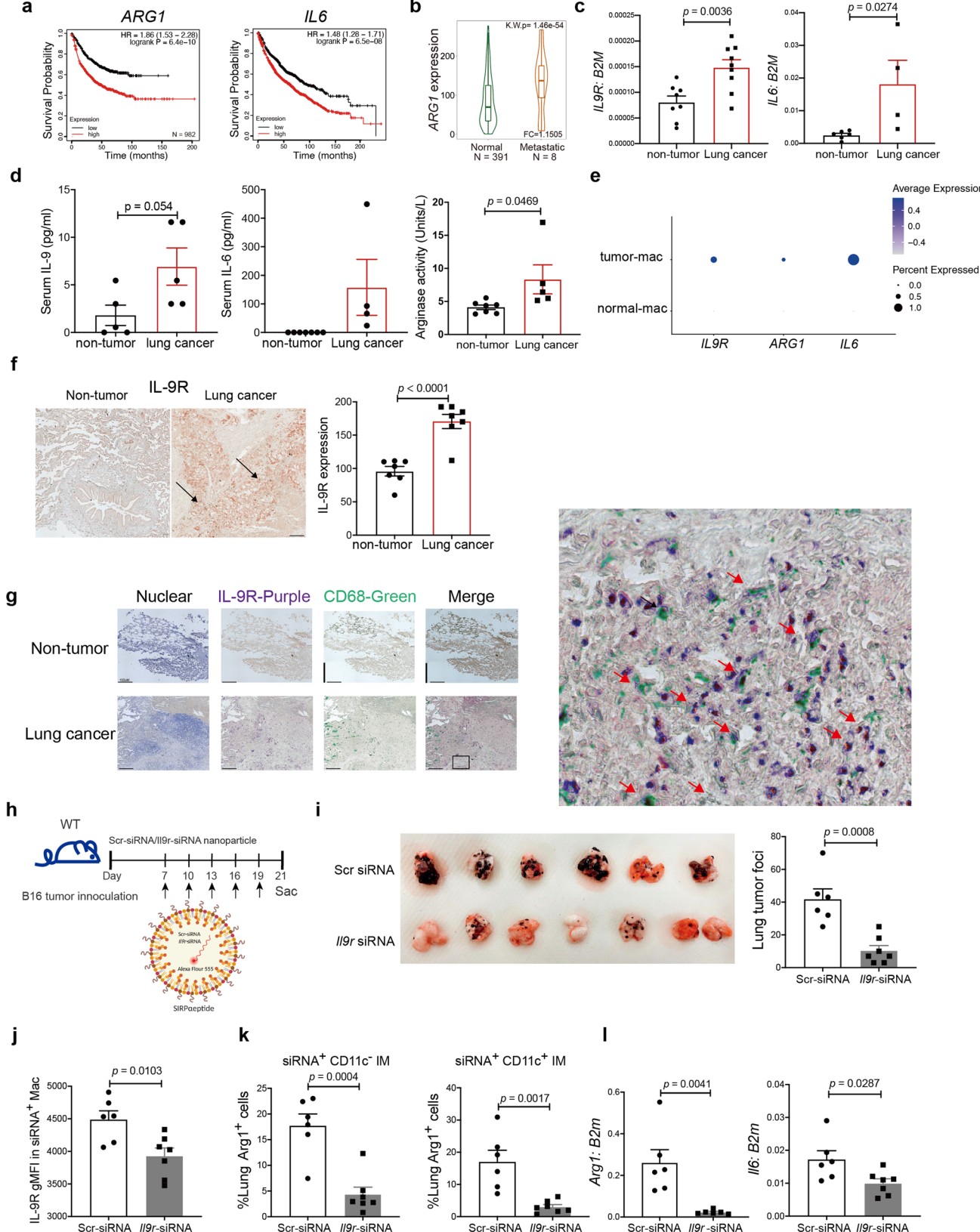

the lung was digested with 1 mg/ml collagenase D at 37 °C for 30 min with rotation. After digestion, the lungs were passed through a mesh and red blood cells lysed with ACK lysis buffer for 3 min (Lonza). The lung cells were resuspended in 5 ml 40% percoll and added into a 15 ml tube. 80% percoll was slowly added to the bottom of the tube. After centrifugation at 325 × g at room temperature for 30 min with no brake, cells at the interface between 40% and 80% percoll was collected and directly passed through a 70 μm cell strainer. Cells were washed with FACS buffer

and kept for FACS analysis. Eosinophils were identified as live Ly6G⁻ SiglecF⁺ CD11c⁻ CD11b⁺ cells; Neutrophils were identified as live Ly6G⁺ CD11b⁺ cells; Mast cells were identified as CD49b⁻ FcεR1⁺ c-Kit⁺ cells; Monocytes were identified as live Ly6c⁺ CD11b⁺ cells; DCs were identified as live MHCII + CD11b⁺ or MHCII⁺ CD103⁺ cells; Total macrophages were identified as live CD64⁺ MerTK⁺ cells; AMs, CD11c⁻ IMs and CD11c⁺ IMs were distinguished by the expression of Siglec-F and CD11c. Blood was collected by Cardiac puncture. A portion of the

**Fig. 9 Therapeutic targeting of IL-9-macrophage axis prevents lung cancer growth. a** Kaplan–Meier plots showing differences in survival among lung cancer patients ($n = 982$ donors) by using data and online tools described in Methods for *ARG1* and *IL6*. **b** Comparison of gene expression in normal lung tissue and metastatic lung tissues by using data and online tool described in the method. The bars represent the proportions of metastatic tumor samples that show higher expression of the selected gene compared to normal samples at each of the quantile cutoff values (minimum, 1st quartile, median, 3rd quartile, maximum). **c** *IL9R* and *IL6* gene expression were analyzed in cells from normal lung tissue and cells in the lung tumor ($n = 8$ donors for non-tumor group and $n = 9$ donors for lung cancer group in left panel, $n = 6$ donors for non-tumor group and $n = 4$ donors for lung cancer group in right panel). **d** Serum IL-9 level, IL-6 level and arginase activity were analyzed from healthy donors and lung cancer patient samples. ($n = 5$ donors for left panel, $n = 7$ donors for non-tumor group and $n = 4$ donors for lung cancer group in middle panel, $n = 7$ donors for non-tumor group and $n = 5$ donors for lung cancer group in right panel). **e** Dot plot showing gene expression in macrophages between normal lung tissue and lung tumor. **f, g** Immunohistochemistry staining of CD68 and IL-9R. Protein expression quantification was performed by using Image J software ($n = 7$ donors), Scale bar $= 100$ µm. **h–l** WT mice were intravenously injected with B16 tumor cell line, 7 days after tumor inoculation, tumor bearing mice were intravenously injected with nanoparticle-siRNA complexes every 72 h. Scr/*Il9r*-siRNA was conjugated with Alexa Flour 555. Nanoparticles were tagged with SIRPα peptide (**h**). **i** Lung tumor growth were analyzed on day 21 ($n = 6$ mice for Scr-siRNA group, $n = 7$ mice for *Il9r*-siRNA group). **j** IL-9R expression in siRNA$^+$ (Alexa Flour 555) lung macrophages were analyzed by flow ($n = 6$ mice for Scr-siRNA group, $n = 7$ mice for *Il9r*-siRNA group). **k** Arg1 production from siRNA$^+$ (Alexa Flour 555) macrophages were analyzed by flow ($n = 6$ mice for Scr-siRNA group, $n = 7$ mice for *Il9r*-siRNA group). **l** siRNA$^+$ (Alexa Flour 555) lung macrophages were sorted by gating on Alexa Flour 555$^+$ MerTK$^+$ CD64$^+$ live cells. Gene expression was analyzed ($n = 6$ mice for groups in left panel, $n = 6$ mice for Scr-siRNA group and $n = 7$ mice for *Il9r*-siRNA group in right panel). Data are the mean ± SEM. Unpaired two-tailed Student *t*-test was used for comparison.

blood was kept for FACS analysis, red blood cells were lysed with ACK lysis buffer for 3 min (Lonza). The remaining blood samples were kept for serum. Serum was taken after centrifugation.

**LLC tumor lung implantation model**. LLC cells ($0.1 \times 10^6$) were resuspended in 50 µl Matrigel and directly implanted into the lung as previously described with modifications[27]. Mice were weighed before the surgery. Mice were anesthetized with isoflurane and Carprofen was subcutaneously injected on the back of the neck. Mice were laid down in the lateral decubitus position, an incision made over the left chest wall, and subcutaneous fat and muscles were spread to expose the 6th and 7th ribs. LLC cells or Matrigel control was injected 3 mm into the lung under direct visualization. The incision site was closed by using surgical glue. Mice were harvested and analyzed after 14 days tumor implantation.

**IL-9 injection model**. Mice were treated with 4 µg IL-9 (Biolegend, 556004) intravenously or intraperitoneally for three days. Mice were euthanized 1 day after final intranasal challenge.

**Arginase activity assay**. For measurements of arginase activity $1 \times 10^6$ cells per sample were harvested and centrifuged at 1000xg at 4 °C for 10 min. Cell pellets were lysed for 10 min in 100 µl of 10 mM Tris-HCL (pH 7.4) containing 1 µM pepstatin A, 1 µM leupeptin, and 0.4% (w/v) Triton X-100. The cell lysate was centrifuged at $14,000 \times g$ at 4 °C for 10 min and use supernatant was assessed for arginase activity using QuantiChrom Arginase Assay Kit (BioAssay Systems) according to the manufacturer's directions.

**Mixed bone marrow chimera**. The F1 generation of CD45.1 X CD45.2 mice were irradiated at 1000 rds. One day after the irradiation, 8 million bone marrow cells (4 million from Boy/J mice, 4 million from *Il9r*$^{-/-}$ mice) were injected to the recipient mice. Mice were used 12 weeks after donor cell injection.

**Macrophage transfer**. Lung macrophage populations were sorted from tumor bearing mice. Cells were transferred into *Il9r*$^{-/-}$ mice (0.3 million from Boy/J (CD45.1$^+$) mice, 0.15 million from YARG mice). Recipient mice were harvested as indicated in the figures.

**In vivo monocyte/macrophage labeling**. Mice were challenged with 150 µl clodronate-containing liposomes (Liposoma, c-005) intravenously followed by 250 µl of fluorescent microspheres (Polysciences, 17154-10) intravenously injected 16–18 h later. GFP$^+$ monocytes were subsequently purified and 1 million cells were injected into recipient mice.

**In vivo siRNA knockdown of *Il9r***. Mice were intravenously injected with B16 tumor cells, 4 days after tumor injection, nanoparticles containing 50 µg control or *Il9r* siRNA (Ambion in vivo siRNA) were injected intravenously in the tumor-bearing mice every 72 h. Targeted liposomal nanoparticles were prepared using the extrusion method detailed in previous reports[32,33,55]. All lipid, and lipid-conjugate components were individually prepared and purified, then combined at the desired stoichiometric ratios in chloroform and dried to form a lipid film. Lipid films were then hydrated siRNA containing buffer and extruded to form the liposomes. A SIRPα-targeting peptide was used to target tumor-associated macrophage and its sequence was reported in Rodriguez et al.[56].

**Mouse BM derived macrophage cell polarization**. Bone marrow cells were isolated and red blood cells were lysed by using ACK buffer. 0.5 million/ml cells were cultured with complement DMEM media containing 20% L929 supernatant (Cell biologics, 3368) for 7 days. IFN-γ (50 ng/ml, Peprotech) for M1 macrophages or IL-4 (20 ng/ml, Peprotech) for M2 macrophages were added on day 7 for 24 h.

**Human macrophage polarization**. De-identified buffy coat blood packs from male healthy donors were purchased from Indiana Blood Center. Peripheral blood mononuclear cells (PBMCs) were isolated by density gradient centrifugation using Ficoll-Paque (GE Healthcare). Buffy coat cells (10 ml) was diluted with 10 ml DPBS and gently added to 15 ml Ficoll-Paque. After spinning down at 400 x g for 30 min at room temperature without the brake, the upper layer was removed. The mononuclear cell layer was collected and transferred to a new conical tube and filled with MACS up to 50 ml. After mixing, cells were centrifuged at $300 \times g$ for 10 min, repeating this washing step three times. Human CD14$^+$ monocytes were isolated from the PBMCs by using magnetic separation (Miltenyi Biotec). Cells were cultured in the presence of GM-CSF (50 ng/ml, Peprotech) or M-CSF (50 ng/ml, Peprotech) for differentiating to M1 or M2 macrophages for 7 days. IFN-γ (20 ng/ ml, Peprotech) or IL-4 (20 ng/ml, Peprotech) was added for 48 h. Cells were harvested for further analysis.

**Real-time quantitative PCR analysis**. RNA was extracted by using TRIzol reagent (ThermoFisher Schientific) or RNeasy Plus Micro Kit (QIAGEN). cDNA synthesis was performed according to manufacturer's instructions (qScript™ cDNA Synthesis Kits, Quantabio). Taqman real time PCR assays (ThermoFisher Scientific) were used for detecting gene expression (Supplementary Table 5). The relative mRNA expression was normalized to housekeeping gene expression (β2-microglobulin).

**Flow cytometry**. Single cell suspensions were stained with a fixable viability dye (eBioscience) and antibodies for surface markers for 30 min at 4 °C, before fixation with 4% formaldehyde for 10 min dark at room temperature. After fixation, cells were permeabilized with permeabilization buffer (eBioscience) for 30 min at 4 °C and stained for cytokines for another 30 min at 4 °C. For transcription factor staining, after surface staining, cells were fixed with Fixation & Permeabilization Buffer (eBioscience) for 2 h or overnight at 4 °C, and then permeabilized with permeabilization buffer (eBioscience). After intracellular staining, cells were washed with FACS buffer and analyzed by LSR4 or Fortessa (BD Biosciences) and analyzed with Flowjo 10.7.1 software (Tree Star). Gating strategies are shown in Supplementary Fig. 7.

**Cell sorting**. Mouse macrophage were stained with anti-CD64, anti-Mertk, anti-CD11c and anti-Siglec-F antibody and viability dye and further sorted with FACSAria or SORPAria (BD Bioscience) by gating on live cells. Human patient samples were stained with viability dye, anti-hCD14 antibody, followed by FACS sorting. Sorted cells were used for further experiments. Details of antibodies are listed in Supplementary Table 4.

**Enzyme linked immunosorbent assay**. IL-9 (Biolegend) and IL-6 (Biolegend) ELISA were performed according to the manufacturer's instruction. Briefly, 96 well-plate were coated with coating antibody overnight at 4 °C. After washing 3 times with the wash buffer, 300 µl ELISA buffer was added to the plate and incubate at room temperature for 2 h. After washing 3 times with washing buffer, 100 µl samples were added to the plate and incubated at room temperature for 2 h. After washing three times, 100 µl diluted detection antibody was added to the plate

and incubated at room temperature for 1 h. After washing the plate three times, 100 μl of diluted Avidin-HRP solution was added to the plate and incubated at room temperature for 30 min at dark. After washing the plate for 3 times, 100 μl substrate was added to the plate. Plates were read at absorbance 450 and 570 nm.

**Multiplex immunohistochemistry.** Lung cancer sections were purchased from OriGene, the detailed information for the patients were listed in Table S5. Sections were placed in a 60 °C heat chamber for 30 min. FFPE tissue sections (5 mm) were deparaffinized in xylene and rehydrated in decreasing concentrations of ethanol (100, 90, 70, and 50% and distilled water; 5 min each). Slides were stained by hematoxylin (S3301, Dako) for 1 min, mounted with TBST buffer (0.1M TRIS-HCl, pH 7.5, 0.15 M NaCl plus 0.05% Tween-20), and coverslipped with Signature Series Cover Glass (12460S, Thermo Scientific), followed by whole tissue scanning using a microscopy (Leica Biosystems) at 20x magnification. After imaging, the staining was removed in organic solvent (50% ethanol, 2 min; 100% ethanol, 2 min; 100% xylene; 2 min, 100% ethanol, 2 min; and 50% ethanol, 2 min). One round of antigen retrieval was then performed, consisting of three cycles of 5 min (high power) in a commercial microwave in citrate buffer (pH 6.0, Sigma, C9999-100ml), followed by cooling to room temperature. Wash with water. Endogenous peroxidase was quenched in 3% hydrogen peroxide for 10 min, followed by rinsing in TBST. Tissue was then blocked in 10% normal goat blocking serum (Thermo Fisher, 50197z) for 10 min, followed by primary antibody incubation overnight in 4 °C. The following day, tissue sections were washed for three times in TBST for 5 min. After washing, sections were incubated with secondary antibody (Nacalai USA Inc., Histofine Simple Stain MAX PO) for 30 min at room temperature. After three times washing, sections were incubated with AEC subtract (Vector laboratories, SK-4200). After 5 min washing in water, sections were covered and imaging as described above. The detailed information of the antibody is listed in Supplementary Table 4.

**Immunofluorescence staining.** Lung sections (5 μm) were washed with PBST three times and blocked with CAS block (Life Technologies, Cat#: 008120). After blocking, sections were incubated with anti-CD31-AF594 antibody (BioLegend) at 4 °C overnight. After washing three times, sections were mounted onto slides with DAPI Fluoromount-G (Southern Biotech). Images were captured using a Nikon Eclipse 80 microscope equipped with Nikon Intensilight epifluorescence and a Nikon DS-Ti3 High speed color sCMDS camera.

**In vitro tumor stimulation.** B16 cells were cultured in the presence of IGF (50 ng/ml, Thermofisher), IL-9 (40 ng/ml, BioLegend,), IL-6 (20 ng/ml, Peprotech) or PBS for 60 min at 37 °C. Cells were harvested and washed with complete DMEM media, followed by fixation with 1% paraformaldehyde for 10 min in the dark at room temperature. After centrifugation at $630 \times g$ for 5 min, cells were permeabilized with ice-cold methanol overnight at $-20$ °C. The following day, cells were centrifuged at $630 \times g$ for 5 min at 4 °C and washed twice with PBS. Cells were then stained with pSTAT3 antibody (BioLegend) or isotype (BD Biosciences) for 30 min at 4 °C. After intracellular staining, cells were washed, resuspended in PBS and analyzed.

**Bulk RNA-Seq.** Lung macrophages were isolated from B16 tumor mice by FACS sorting. RNA was isolated by using RNease micro kit (Qiagen). Purified total RNA was first evaluated for its quantity, and quality, using Agilent Bioanalyzer 2100. A RIN (RNA Integrity Number) of seven or higher was required to pass the quality control. One nanogram of total RNA per sample were used for library preparation. cDNA was first synthesized using SMART-Seq v4 Ultra Low Input RNA Kit for Sequencing (Takara Clontech Laboratories, Inc.). Dual indexed cDNA library was then prepared using Nextera XT DNA Library Prep Kit (Illumina, Inc.). Each library was quantified and its quality assessed by Qubit and Agilent Bioanalyzer, and multiple libraries were pooled in equal molarity. Average size of library insert was about 250–300bp. The pooled libraries were then denatured, neutralized before loading to NovaSeq 6000 sequencer for 100 bp paired-end sequencing (Illumina, Inc.). Approximately 30–40M reads per library was generated. A Phred quality score (Q score) was used to measure the quality of sequencing. More than 95% of the sequencing reads reached Q30 (99.9% base call accuracy).

**RNA-Seq analysis.** The sequencing data were first assessed using FastQC (v.0.11.5, Babraham Bioinformatics, Cambridge, UK) for quality control. All sequenced libraries were mapped to the mouse genome (UCSC mm10) using STAR RNA-seq aligner (v.2.5)[57] with the following parameter: "--outSAMmapqUnique 60". The read distribution across the genome was assessed using bamutils (from ngsutils v.0.5.9)[58]. Uniquely mapped sequencing reads were assigned to mm10 refGene genes using featureCounts (subread v.1.5.1)[59] with the following parameters: "-s 2 –p –Q 10". Each patient was analyzed independently and genes with read count per million (CPM) < 0.5 in more than the number of sample replicates in one group were removed from the comparisons. The data was normalized using TMM (trimmed mean of M values) method. Multi-dimensional scaling analysis was done with limma (v.3.38.3)[60]. Differential expression analysis was performed using edgeR (v.3.28.1)[61,62]. False discovery rate (FDR) was computed from $p$ values using the Benjamini-Hochberg procedure. GSEA v2[63,64] was used to test for gene enrichment of clusters with all gene sets from the Molecular Signature Database (MSigDB v5.2).

**Analysis of single cell RNA-Seq.** Processed lung cancer patient single cell RNA-Seq data were download from dataset under GSE154826[22]. The data was further analysis with the R package Seurat[65,66] (Seurat 3.1.1) with Rstudio version 1.2.5001 and R version 3.5.1. Cells with low number of detected genes/UMIs and high mitochondrial gene content were excluded. For gene expression data analysis, gene expression levels for each cell were log normalized with the NormalizeData function in Seurat. Highly variable genes were subsequently identified with Find-VariableFeatures using the "vst" approach. To integrate the single cell data of the normal tissue samples and tumor samples, functions FindIntegrationAnchors and IntegrateData from Seurat were applied. The integrated data was scaled and PCA was performed. Clusters were identified with the Seurat functions FindNeighbors and FindClusters. The FindConservedMarkers function was subsequently used to identify cell cluster specific marker genes. Cell cluster identities were manually defined with the cluster-specific marker genes or known marker genes. The cell clusters were visualized using the t-Distributed Stochastic Neighbor Embedding (t-SNE) plots and Uniform Manifold Approximation and Projection (UMAP) plots. Human TAM genes showed in Fig. 4e were genes expressed in human monocytes and macrophages from lung cancer patient tumor site. Mouse IM genes are genes expressed in WT macrophage isolated from mouse B16 tumor models.

**Lung cancer survival curves and gene expression.** Patient data for lung cancer survival was defined based on published datasets[20] and analysis at www.kmplot.com/lung. IL9, IL9R and ARG1 expression in metastatic tumors was defined using TNMplot.com.

**Histology.** Lung tissue was fixed with 4% formalin for 24 h at room temperature. Tissues were embedded in paraffin, sectioned, and further stained with H&E or periodic acid-Schiff (PAS) stain.

**Statistics analysis.** Statistical was analyzed by using GraphPad Prism V_8.0 and V_7.0 (GraphPad Software) and presented as means ± SEM. Unpaired or paired Student t tests and one-way or two-way ANOVA analysis were used in data analysis. A $p$ value < 0.05 was considered statistically significant.

## Data availability

The RNA-seq data generated in this study have been deposited in the Gene Expression Omnibus database under accession code GSE174005. Human lung cancer patient single cell RNA-Seq data were downloaded from Gene Expression Omnibus database GSE154826. The information of mouse genome (mm10) is available on UCSC Genome Browser (http://genome.ucsc.edu). The raw numbers for charts and graphs are available in the Source Data file whenever possible. Source data are provided with this paper.

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

## Author contributions

Y.F. and M.H.K. designed the experiments, Y.F. performed the experiments, analyzed the data. Y.F. performed the bioinformatic experiments and analysis. A.P., J.W., B.Z., A.C., C.L.C., J.Z., H.Z., A.J.F., S.K., D.O., and L.H. performed and assisted with the experiments. J.C.R., S.P., H.G. Y.L., L.Y., R.M.T., P.L.L., R.A.F., S.T., D.K., J.S., B.B., C.R.S., and K.Y. provided reagents, mice and advice. M.H.K. coordinated the project. Y.F. and M.H.K. wrote the manuscript with input from all authors. The funding agencies were not involved in conceptualization, completion or publication of the studies.

## Competing interests

D.T.O., S.K., B.B., Y.F., and M.H.K. are authors on pending patent applications for nanoparticle approaches. R.A.F. is an advisor to Glaxo Smith Kline, Zai Lab, and Ventus Therapeutics though not in the context of this work. The remaining authors declare no competing interests.
