## [Peer Review File · Nature Communications]

Mouse pulmonary interstitial macrophages mediate the pro-tumorigenic effects of IL-9REVIEWER'S COMMENTS

Reviewer #1 (Remarks to the Author):

The manuscript by Fu et al., investigates the role of IL-9 in lung cancer. The authors used a combination of human and murine experiments to demonstrate that IL-9 triggers the expansion of CD11c⁺ interstitial macrophages that react to IL-9 and promote tumor growth. Mechanistically, the authors identified IL9 signaling in macrophages induces arginase1-mediated tumor growth. The experiments are elegant and well executed. The conceptual framework of the study is very interesting and clinically relevant, there are few points that need to be addressed to strengthen the central hypothesis of the current study.

Specific comments:

- 1) The vast majority of the results is a snapshot of the lungs at 21 days post-B16 cells injection, where there are less tumor foci in the lungs of IL9^{-/-} and IL9R^{-/-}. Thus, it will be important to conduct kinetic experiment to assess the dynamics of the macrophage populations and other immune cells over the course of tumor development. In addition to the frequency, please also provide absolute numbers (that can be normalized to the g of tissue). Additionally, the authors also need to provide the frequency and absolute numbers of AM and IMs in naïve animals for all the strains used in this study. Moreover, the potential contribution of IL-9 signaling during the early phase/innate immunity of tumor growth versus the late phase/adaptive immunity should be better discussed.
- 2) A recent study by the Merad's group analysed the lung macrophage landscape in human and murine lung cancer and demonstrated that AM play a detrimental role in early cancer progression (Casanova-Acebes et al., Nature 2021). This article should be cited in the current manuscript. In addition, AM express the IL9 receptor at similar levels to IM (Fig. 1f). Is it possible that early IL9 signaling in AM was the main driver of tumor progression in WT mice? This can be answered in the adoptively transferred model of AM, as has been done in figure 5d.
- 3) What is the origin of the CD11c⁺ IM? Are they locally proliferating or recruited from the bone-marrow?
- 4) While the experiments support that IM play a role in tumor development, the authors have not addressed whether this is the direct effect of IL-9 on IM or an indirect effect of IL-9 on tumor microenvironment that could reprogram the macrophages. The use of a conditional IL9R knockout mice for IM will answer this question. Alternatively, the authors can adoptively transfer IL-9-stimulated WT-macrophages into IL9 KO mice and then assess tumor development? Additionally, the authors could also evaluate whether IL-9 directly induce Arginase 1 expression in macrophages.
- 5) The gating strategy to identify the macrophages populations in the lungs need some refinement. The reviewer agrees that MerTK and CD64 are necessary to identify the total lung macrophage populations (Gibbins et al. AJRCMB 2017, Chakarov et al., Science 2020). Then IM and AM are separated first based on the expression of SiglecF and CD11b, to identify AM (SiglecF⁺ CD11b⁻) and IM (SiglecF⁻ CD11b⁺). IM are further subgated based on the expression of MHC-II and Lyve-I to identify IM1 and IM2 (Gibbins et al. AJRCMB 2017, Chakarov et al., Science 2020), which have different localization and functions. The authors should use that gating strategy in one key experiment to better characterize the CD11c⁺/⁻ population.

Reviewer #2 (Remarks to the Author):

Fu et al., utilize preclinical models of lung primary and metastatic lung cancer to show the protumoral role of IL9 signaling. By analyzing previously available scRNAseq and bulk RNAseq data, they found that IL9/IL9R signaling correlates with reduced survival in lung cancer patients. Further exploration of the IL9-IL9R axis led the authors to establish macrophages as the main responder population. The authors found that IL9-mediated signaling regulates macrophage balance in the lung, expanding TAMs (mainly CD11c⁺ and CD11c^{NEG} IMs) while reducing the pool of resident AMs. Utilizing bulk RNAseq the authors found that some immunosuppressive molecules are also expressed by IMs in lung tumors. Mechanistically, IL9 mediated signaling on IM triggers Arg1 and IL6 expression which feeds tumor growth. Therapeutic targeting of IL9R using siRNA nanoparticles restricted Arg1⁺ expressing macrophages and reduced tumor growth. While the study is interesting and experiments are well-performed, some conclusions should be further supported by experimental data. Hence the authors should address:

1. Lines 89/92/163: authors claim that IL9R signaling affects macrophage heterogeneity in multiple mouse models. However, the authors only show the presence of the main macrophage populations (AMs and 2 classes of IMs). Several studies have now shown that macrophage heterogeneity in lung tumors is way more complex than 3 subsets (see Zilionis et al., *Immunity* 2019, Casanova-Acebes et al., *Nature* 2021). Hence the statement related to IL9 signaling inducing alterations in macs heterogeneity should be restricted to the populations analyzed. Importantly, the authors only analyzed relative % and not absolute numbers, which might reflect alterations in other populations. Could the authors provide absolute numbers of IMs and AMs in the different models utilized?
2. As shown in Supp Figure 1h CD4 T cells are the main producers of IL9. Did the authors try to deplete CD4 T cells to see whether CD4-derived IL9 is sufficient to promote IM expansion?
3. To formally show that IL9R signaling on IM triggers downstream Arg1 signaling authors should culture FACS sorted IMs from B16 tumors (both CD11c⁺ and CD11c⁻) +/- recombinant IL9 and measure Arg1 expression by FACS.
4. What is the mechanism by which IL9R signaling expands IMs? As shown in Supp. Figure 1i, monocytes are significantly expanded in the bone marrow, hence an increased myelopoiesis in IL9R-deficient mice might explain an increased number of circulating monocytes acting as precursors of IMs. Authors should quantify monocyte numbers in blood and lung-tumoral models. Conversely, what is the mechanism that favors AM expansion upon IL9R deficiency: are those AMs proliferating?
5. Line 189-190: authors found a high overlap between transcriptional signatures of human and murine IM-TAMs (Figure 4e). This is quite surprising since as reported by Zilionis et al., TAMs between murine and human species showed minor overlap. Could the authors state how this analysis was carried out in the methods? Could the authors comment on the discrepancies between both studies?
6. Immunohistochemistry Figure 9g: There is no co-expression of IL9R with CD68 and Arg1. Could the authors explain this or choose a representative image that supports their previous observations in human lung cancer? Insets do not show clear staining for CD68.

Minor points: - Figure 3. Authors should carefully state in which tumor model DEG expression of TAMs was carried out. - Given the absence of changes in CD4, CD8 % of cells and effector molecules (TNF α , granzyme, IFN γ) and the profound effect in tumor clearance, did the authors check whether DC numbers were increased in the absence of IL9R signaling? - Did siRNA nanoparticles affect the ratio of macrophages in WT-Scr vs IL9R-siRNA?

Reviewer #3 (Remarks to the Author):

Manuscript number: NCOMMS-21-28046

Title: "Pulmonary interstitial macrophages are required to mediate the pro-tumorigenic effects of IL-9" by Yongyao Fu and colleagues.

In the present manuscript the authors provide a comprehensive study on the role of Interleukin-(IL-9) in lung cancer and the role of macrophages in promoting IL-9-mediated tumour growth.

In detail the authors provide whole set of experimental approaches and strong experimental data to demonstrate that pulmonary interstitial macrophages (IM) are IL-9-responding cells serving as pro-tumorigenic tumour-associated macrophages (TAMs) in mice and humans.

Employing transcriptomic analyses of wildtype and Il9r-deficient IM the authors identify genes associated with an alternatively activated macrophage phenotype (e.g. Arg1, Vegfa) to be involved in the molecular mechanisms of IL-9-dependent IM polarization. Further analyses employing Arg1 deficient macrophages the authors demonstrate a direct role of IL-9-induced Arg1 and Il6 in IM-mediated lung tumour growth.

This is a well-written manuscript with a lot of sophisticated experimental work providing strong evidence for a role of IM in IL-9-mediated lung tumour growth. However, some concerns rose during the review:

Major critique:

1. Employing sophisticated experimental approaches and using Il9r deficient and wildtype mouse strains the authors identify IM as an important cell type responding to IL-9 and promoting tumour growth. Given that IL-9 signalling has direct pro-tumorigenic effects in multiple cancers the authors should be encouraged to analyse IL-9 receptor expression in the tumour cell lines used and its direct effect on tumour cells and tumour growth. For example, what is the effect of IL-9 in absence of macrophages analogous to the experiments presented in Supp Fig 2e-f?
2. In Figure 6 the authors nicely demonstrate that Il9r deficiency results in reduced expression of a set of genes that is associated with an alternatively activated macrophage phenotype and angiogenesis. Does IL-9 affect angiogenesis in the preclinical lung cancer models employed in an IM-dependent manner?
3. In Figures 4 and 5 the authors demonstrate that Il9r deficiency in IM results in strongly attenuated tumour growth. What is the impact of Il9r deficiency in IM on the adaptive anti-tumour immune response? A more detailed analysis than the one that is given e.g. in Supp Fig 2a-b would be desirable.
4. In Figure 8 the authors demonstrate that IL-9 induces IL-6 expression in Arg1-expressing IM. Does IM-produced IL-6 has a direct effect on STAT3 phosphorylation and proliferation of tumour cells or is this effect indirect by acting on e.g. antigen-presenting cells like dendritic cells?

Minor critique:

1. In Figure 3 the authors demonstrate results of the RNA-Seq analysis on CD11c-negative and CD11c-positive IM. It would be interesting to compare these results with the RNA-Seq analysis of IMs from wildtype and Il9r-deficient mice (Fig. 6) in order to further understand IL-9 receptor-dependent gene regulation in these IM subsets.
2. In legend of Figure 4 the description of 4b and 4c seems to be interchanged.

Reviewer #1 (Remarks to the Author):

The manuscript by Fu et al., investigates the role of IL-9 in lung cancer. The authors used a combination of human and murine experiments to demonstrate that IL-9 triggers the expansion of CD11c+ interstitial macrophages that react to IL-9 and promote tumor growth. Mechanistically, the authors identified IL9 signaling in macrophages induces arginase1-mediated tumor growth. The experiments are elegant and well executed. The conceptual framework of the study is very interesting and clinically relevant, there are few points that need to be addressed to strengthen the central hypothesis of the current study.

Specific comments:

1) The vast majority of the results is a snapshot of the lungs at 21 days post-B16 cells injection, where there are less tumor foci in the lungs of Il9-/- and Il9R-/. Thus, it will be important to conduct kinetic experiment to assess the dynamics of the macrophage populations and other immune cells over the course of tumor development. In addition to the frequency, please also provide absolute numbers (that can be normalized to the g of tissue). Additionally, the authors also need to provide the frequency and absolute numbers of AM and IMs in naïve animals for all the strains used in this study. Moreover, the potential contribution of IL-9 signaling during the early phase/innate immunity of tumor growth versus the late phase/adaptive immunity should be better discussed.

We thank the reviewer for the positive comments.

In response to the first comment we have added a time course analysis of tumor and macrophages that has included absolute numbers as well. This data is in Supplementary Fig. 1.

We have added further discussion on how IL-9 may be important for early and late stages of tumor growth.

2) A recent study by the Merad's group analysed the lung macrophage landscape in human and murine lung cancer and demonstrated that AM play a detrimental role in early cancer progression (Casanova-

Mark H. Kaplan, PhD Nicole Brown Chair, Department of Microbiology and Immunology
Director of Basic Sciences, Brown Center for Immunotherapy
635 Barnhill Drive, MS 420 Indianapolis, IN 46202 317-278-3696 mkaplan2@iu.edu

Acebes et al., Nature 2021). This article should be cited in the current manuscript. In addition, AM express the IL9 receptor at similar levels to IM (Fig. 1f). Is it possible that early IL9 signaling in AM was the main driver of tumor progression in WT mice? This can be answered in the adoptively transferred model of AM, as has been done in figure 5d.

We have cited and added discussion on the report by Merad and colleagues. We have added data showing that adoptive transfer of AMs results in enhanced tumor growth to Fig. 5. However, we feel that this is likely due to the immunosuppressive functions of AMs and not due to IL-9 signaling because the adoptive transfer increases the overall number of AMs which are not deficient in the *Il9r*^{-/-} mice (Figure. 2, Supplementary Figures 1 and 2). IMs are more likely the relevant IL-9 responders because TAMs, at the time point of maximal differences between WT and *Il9r*^{-/-} mice, have an IM phenotype (Fig. 4). Casanova-Acebes et al also found that IMs become dominant TAMs later in tumor development. Moreover, IMs have greater expression of Arg1, a critical component downstream of IL-9 (Fig. 7A). We have added discussion on these points as well.

3) What is the origin of the CD11c+ IM? Are they locally proliferating or recruited from the bone-marrow?

We have performed several experiments to address this and results indicate it is a combination of both mechanisms. The mixed bone marrow chimera experiment (Fig.4) suggested bone marrow monocytes can develop into CD11c+IM. WT bone marrow monocytes showed higher proliferative capacity than *Il9r*^{-/-} bone marrow monocytes, which indicates IL-9 signaling could promote CD11c+ IMs by expanding the bone marrow monocyte population, although the numbers of monocytes was not significantly different (Supplementary Fig. 2c-d). We did not see different proliferation of blood and lung monocytes (Supplementary Fig 2c-d). We added an experiment to Fig. 4 where bead-labeled monocytes are injected. The ratio of bead-labeled to -unlabeled CD11c+ IMs is about 1:1, suggesting that both mechanisms contribute to the expansion of these cells.

4) While the experiments support that IM play a role in tumor development, the authors have not addressed whether this is the direct effect of IL-9 on IM or an indirect effect of IL-9 on tumor microenvironment that could reprogram the macrophages. The use of a conditional IL9R knockout mice for IM will answer this question. Alternatively, the authors can adoptively transfer IL-9-stimulated WT-macrophages into IL9 KO mice and then assess tumor development? Additionally, the authors could also evaluate whether IL-9 directly induce Arginase 1 expression in macrophages.

There are several experiments that indicate a direct effect on macrophages. First, in the adoptive transfer experiments, wild type monocytes (as a source of lung macrophages) or lung macrophages are transferred to *Il9r*^{-/-} mice (Fig. 4i-k and 5). In this system, macrophages are the only IL-9 responsive cells in the lung that mediate the effect. Moreover, in the mixed bone marrow chimera experiment, we show that there are similar changes to the macrophage populations, demonstrating that the effects of IL-9 are intrinsic to the macrophages themselves. Finally, we have performed the last experiment suggested by the reviewer; we have isolated macrophages and stimulated with IL-9 in vitro to further support a direct effect (Fig. 7e). Unfortunately, the other experiment suggested by the reviewer is not possible as a conditional IL-9R mutant mouse is not available.

5) The gating strategy to identify the macrophages populations in the lungs need some refinement. The reviewer agrees that MerTK and CD64 are necessary to identify the total lung macrophage populations (Gibbins et al. AJRCMB 2017, Chakarov et al., Science 2020). Then IM and AM are separated first based on the expression of SiglecF and CD11b, to identify AM (SiglecF+ CD11b-) and IM (SiglecF- CD11b+). IM are further subgated based on the expression of MHC-II and Lyve-1 to identify IM1 and IM2 (Gibbins et al. AJRCMB 2017, Chakarov et al., Science 2020), which have different localization and functions. The authors should use that gating strategy in one key experiment to better characterize the CD11c+/- population.

We have added details on our gating strategy to Supplementary Figure 1. We have added analysis to better define the IM subpopulations in Supplementary Fig. 1 and demonstrate that the IM3 population is most affected by loss of IL-9 signaling. We also show that Lyve-1 expression is not affected by IL-9R signaling, but that IM3 cells are Lyve-1+, which contrasts Gibbings et al. This may be a result of the tumor development, with previous studies being performed under homeostatic conditions, and this is mentioned in the text.

Reviewer 2

Fu et al., utilize preclinical models of lung primary and metastatic lung cancer to show the pro- tumoral role of IL9 signaling. By analyzing previously available scRNAseq and bulk RNAseq data, they found that IL9/IL9R signaling correlates with reduced survival in lung cancer patients. Further exploration of the IL9-IL9R axis led the authors to establish macrophages as the main responder population. The authors found that IL9-mediated signaling regulates macrophage balance in the lung, expanding TAMs (mainly CD11c+ and CD11c^{NEG} IMs) while reducing the pool of resident AMs. Utilizing bulk RNAseq the authors found that some immunosuppressive molecules are also expressed by IMs in lung tumors. Mechanistically, IL9 mediated signaling on IM triggers Arg1 and IL6 expression which feeds tumor growth. Therapeutic targeting of IL9R using siRNA nanoparticles restricted Arg1+ expressing macrophages and reduced tumor growth. While the study is interesting and experiments are well-performed, some conclusions should be further supported by experimental data. Hence the authors should address:

1. *Lines 89/92/163: authors claim that IL9R signaling affects macrophage heterogeneity in multiple mouse models. However, the authors only show the presence of the main macrophage populations (AMs and 2 classes of IMs). Several studies have now shown that macrophage heterogeneity in lung tumors is way more complex than 3 subsets (see Zilionis et al., Immunity 2019, Casanova-Acebes et al., Nature 2021). Hence the statement related to IL9 signaling inducing alterations in macs heterogeneity should be restricted to the populations analyzed. Importantly, the authors only analyzed relative % and not absolute numbers, which might reflect alterations in other populations. Could the authors provide absolute numbers of IMs and AMs in the different models utilized?*

We thank the reviewer for these points. We have added absolute cell numbers for key experiments in Supplementary Figure 1. We appreciate the point on heterogeneity and it is clear that populations can be divided by various markers. Reviewer 1 also made this point and we have added data. Casanova-Acebes defined macrophages primarily in two populations (the others being monocytes) that correspond to alveolar or interstitial phenotype cells. We have added discussion on this point as well. We have also limited the use of the term ‘heterogeneity’ and tried to describe the results more accurately.

2. *As shown in Supp Figure 1h CD4 T cells are the main producers of IL9. Did the authors try to deplete CD4 T cells to see whether CD4-derived IL9 is sufficient to promote IM expansion?*

The reviewer raises an important point. We didn’t perform the CD4 depletion because CD4+ cells are likely contributing more than IL-9 to the environment and interpreting the experiment could be difficult. Instead, we utilized a new *Il9* conditional mutant mouse we have generated and demonstrated that when IL-9 is depleted in T cells there is a similar alteration of macrophage populations to that observed when IL-9R is deleted. These data are in Fig. 2. As a further confirmation for this point, we have added data that mast cells (another IL-9 producing cell in Supplementary Fig. 1o) are not required for tumor growth and mast cell deficient mice do not have the same changes in macrophage populations observed in *Il9r-/-* mice. Data with mast cell deficient mice is shown in Supplementary Fig. 1p-q.

3. *To formally show that IL9R signaling on IM triggers downstream Arg1 signaling authors should culture FACS sorted IMs from B16 tumors (both CD11c+ and CD11c-) +/- recombinant IL9 and measure Arg1 expression by FACS.*

We agree this is an important point also raised by Reviewer 1 and we have added that experiment to Fig. 7e.

4. *What is the mechanism by which IL9R signaling expands IMs? As shown in Supp. Figure 1i, monocytes are significantly expanded in the bone marrow, hence an increased myelopoiesis in IL9R-deficient mice might explain an increased number of circulating monocytes acting as precursors of IMs. Authors should quantify monocyte numbers in blood and lung-tumoral models. Conversely, what is the mechanism that favors AM expansion upon IL9R deficiency: are those AMs proliferating?*

We have performed several experiments to address this and results indicate it is a combination of both mechanisms. The mixed bone marrow chimera experiment (Fig. 4) suggested bone marrow monocytes can develop into CD11c+IM. WT bone marrow monocytes showed higher proliferation capacity than Il9r-/- bone marrow monocytes, which indicates IL-9 signaling could promote CD11c+ IMs by expanding the bone marrow monocyte population (Supplementary Fig. 2c-d). We did not see differences in proliferation of blood and lung monocytes (Supplementary Fig. 2c-d). We have added bone marrow monocyte and lung monocyte numbers in Supplementary Fig. 2c-d. We added an experiment to Fig. 4i-k where bead-labeled monocytes are injected. The ratio of bead-labeled to -unlabeled CD11c+ IMs is about 1:1, suggesting that both mechanisms contribute to the expansion of these cells.

Comparing to WT-PBS mice, AM in WT-B16 tumor mice showed less proliferation marker expression. And there is no difference of Ki67 expression between Il9r-/- PBS vs Il9r-/- B16 mice. So IL-9 signaling may inhibit the proliferation of AMs in lung tumor environment.

5. *Line 189-190: authors found a high overlap between transcriptional signatures of human and murine IM-TAMs (Figure 4e). This is quite surprising since as reported by Zilionis et al., TAMs between murine and human species showed minor overlap. Could the authors state how this analysis was carried out in the methods? Could the authors comment on the discrepancies between both studies?*

We have added a description of our analysis and we have also added a discussion on these points. While Zilionis et al described macrophages as being the most divergent among the populations in lung cancer, Casanova-Acebes et al (Fig. 1) identified many parallels in patient samples and their mouse model. We observe many parallels between our AM and IM populations with those that Casanova-Acebes identified in what they termed TRM and MDM populations including the differential expression of Arg1. We have made these points in the revised text.

6. *Immunohistochemistry Figure 9g: There is no co-expression of IL9R with CD68 and Arg1. Could the authors explain this or choose a representative image that supports their previous observations in human lung cancer? Insets do not show clear staining for CD68.*

We agree that staining was not optimal, particularly the Arg1 staining and we have deleted that component of the analysis. We have added an enlarged image for this figure and added red arrows to indicate cells that have co-expression of CD68 and IL-9R. A lower magnification panel was added to the Supplementary Figures.

Minor points:

- *Figure 3. Authors should carefully state in which tumor model DEG expression of TAMs was carried out.*

We have modified the description that the data are from the B16 melanoma model and apologize for the lack of detail.

- Given the absence of changes in CD4, CD8 % of cells and effector molecules (TNF α , granzyme, IFN γ) and the profound effect in tumor clearance, did the authors check whether DC numbers were increased in the absence of IL9R signaling?

The numbers of DCs are not affected and this is shown in Supplementary Fig. 5. However, there are differences in the activation state of the DCs and that data is also shown in Supplementary Fig. 5.

- Did siRNA nanoparticles affect the ratio of macrophages in WT-Scr vs IL9R-siRNA?

The nanoparticles did have a similar effect on macrophage populations and we have added the data to Supplementary Fig.5.

Reviewer #3 (Remarks to the Author):

Manuscript number: NCOMMS-21-28046

Title: "Pulmonary interstitial macrophages are required to mediate the pro-tumorigenic effects of IL-9" by Yongyao Fu and colleagues.

In the present manuscript the authors provide a comprehensive study on the role of Interleukin-(IL-)9 in lung cancer and the role of macrophages in promoting IL-9-mediated tumour growth.

In detail the authors provide whole set of experimental approaches and strong experimental data to demonstrate that pulmonary interstitial macrophages (IM) are IL-9-responding cells serving as pro-tumorigenic tumour-associated macrophages (TAMs) in mice and humans.

*Employing transcriptomic analyses of wildtype and *Il9r*-deficient IM the authors identify genes associated with an alternatively activated macrophage phenotype (e.g. *Arg1*, *Vegfa*) to be involved in the molecular mechanisms of IL-9-dependent IM polarization. Further analyses employing *Arg1* deficient macrophages the authors demonstrate a direct role of IL-9-induced *Arg1* and *Il6* in IM-mediated lung tumour growth. This is a well-written manuscript with a lot of sophisticated experimental work providing strong evidence for a role of IM in IL-9-mediated lung tumour growth. However, some concerns rose during the review: Major critique:*

*1. Employing sophisticated experimental approaches and using *Il9r* deficient and wildtype mouse strains the authors identify IM as an important cell type responding to IL-9 and promoting tumour growth. Given that IL-9 signalling has direct pro-tumorigenic effects in multiple cancers the authors should be encouraged to analyse IL-9 receptor expression in the tumour cell lines used and its direct effect on tumour cells and tumour growth. For example, what is the effect of IL-9 in absence of macrophages analogous to the experiments presented in Supp Fig 2e-f?*

IL-9R is not expressed on B16 melanoma in Fig.2f or LLC cells (Supplementary Fig. 3a). Culture of either tumor line with IL-9 does not obviously impact cell growth or death (Supplementary Fig. 3b-d).

*2. In Figure 6 the authors nicely demonstrate that *Il9r* deficiency results in reduced expression of a set of genes that is associated with an alternatively activated macrophage phenotype and angiogenesis. Does IL-9 affect angiogenesis in the preclinical lung cancer models employed in an IM-dependent manner?*

We have added data in Fig. 6g-h showing that CD31 staining is decreased in the tumor-bearing *Il9r*^{-/-} mice and that adoptive transfer of wild type macrophages to *Il9r*^{-/-} mice increases CD31⁺ cells.

*3. In Figures 4 and 5 the authors demonstrate that *Il9r* deficiency in IM results in strongly attenuated tumour growth. What is the impact of *Il9r* deficiency in IM on the adaptive anti-tumour immune response? A more detailed analysis than the one that is given e.g. in Supp Fig 2a-b would be desirable.*

We have added data to Supplementary Fig. 5e showing that pSTAT3 is modestly though significantly decreased in DCs from IL-9R-deficient mice. We have further added data that the activation state but not the numbers of DC populations are increased in the absence of IL-9 signaling. Additional data on T cell phenotypes was also added to Supplementary Fig. 4g-i.

4. In Figure 8 the authors demonstrate that IL-9 induces IL-6 expression in Arg1-expressing IM. Does IM-produced IL-6 has a direct effect on STAT3 phosphorylation and proliferation of tumour cells or is this effect indirect by acting on e.g. antigen-presenting cells like dendritic cells?

There is considerable evidence in the literature that STAT3 is constitutively active in B16 melanoma cells. We have added citations of that work. We have also added data to Supplementary Fig. 5c showing that neither IL-6 nor IL-9 activate STAT3 any further.

Minor critique:

1. In Figure 3 the authors demonstrate results of the RNA-Seq analysis on CD11c-negative and CD11c-positive IM. It would be interesting to compare these results with the RNA-Seq analysis of IMs from wildtype and Il9r-deficient mice (Fig. 6) in order to further understand IL-9 receptor-dependent gene regulation in these IM subsets.

We have added a Venn diagram in Fig. 6k to indicate the overlap in genes that are regulated by IL-9 and genes that are enriched in the IM subset.

2. In legend of Figure 4 the description of 4b and 4c seems to be interchanged

This has been fixed.

Once again, we thank the reviewers for their comments that have helped to improve this manuscript. We hope that it is now acceptable for publication in *Nature Communications*.

Sincerely,

Mark H. Kaplan, PhD.

REVIEWERS' COMMENTS

Reviewer #1 (Remarks to the Author):

The authors have effectively addressed my concerns.

Reviewer #2 (Remarks to the Author):

The authors have conveniently answered all the comments raised by the reviewer.

Reviewer #3 (Remarks to the Author):

In the revised manuscript by Mark Kaplan and colleagues, "Pulmonary interstitial macrophages are required to mediate the pro-tumorigenic effects of IL-9" the authors satisfactorily addressed most important points of my concerns. Further concerns have been appropriately addressed in response to the reviewers' concerns as well as in the discussion of the manuscript. Taken together, the authors have further improved their manuscript by appropriately responding to most of the criticism raised by inclusion of additional data further strengthening their conclusions.